

# Using synthetic case studies to explore the spread and calibration of ensemble atmospheric dispersion forecasts

Andrew R. Jones, Susan J. Leadbetter, and Matthew C. Hort

Met Office, Exeter, EX1 3PB, UK

**Correspondence:** Andrew R. Jones (andrew.jones@metoffice.gov.uk)

**Abstract.** Ensemble predictions of atmospheric dispersion that account for the meteorological uncertainties in a weather forecast are constructed by propagating the individual members of an ensemble numerical weather prediction forecast through an atmospheric dispersion model. Two event scenarios involving hypothetical atmospheric releases are considered: a near-surface radiological release from a nuclear power plant accident, and a large eruption of an Icelandic volcano releasing volcanic ash into the upper air. Simulations were run twice-daily in real time over a four month period to create a large data set of cases for this study. Performance of the ensemble predictions is measured against retrospective simulations using analysed meteorological fields. The focus of this paper is on comparing the spread of the ensemble members against forecast errors and on the calibration of probabilistic forecasts derived from the ensemble distribution.

Results show good overall performance by the dispersion ensembles in both studies, but with simulations for the upper air ash release generally performing better than those for the near-surface release of radiological material. The near-surface results demonstrate a sensitivity to the release location, with good performance in areas dominated by the synoptic-scale meteorology and generally poorer performance at some other sites where, we speculate, the global-scale meteorological ensemble used in this study has difficulty in adequately capturing the uncertainty from local and regional scale influences on the boundary layer. The ensemble tends to be under-spread, or over-confident, for the radiological case in general, especially at earlier forecast steps. The limited ensemble size of 18 members may also affect its ability to fully resolve peak values or adequately sample outlier regions. Probability forecasts of threshold exceedances show a reasonable degree of calibration, though the over-confident nature of the ensemble means that it tends to be too keen on using the extreme forecast probabilities.

Ensemble forecasts for the volcanic ash study demonstrate an appropriate degree of spread and are generally well-calibrated, particularly for ash concentration forecasts in the troposphere. The ensemble is slightly over-spread, or under-confident, within the troposphere at the first output time step T+6, thought to be attributable to a known deficiency in the ensemble perturbation scheme in use at the time of this study, but improves with probability forecasts becoming well-calibrated here by the end of the period. Conversely, an increasing tendency towards over-confident forecasts is seen in the stratosphere, which again mirrors an expectation for ensemble spread to fall away at higher altitudes in the met ensemble. Results in the volcanic ash case are also broadly similar between the three different eruption scenarios considered in the study, suggesting that good ensemble performance might apply to a wide range of eruptions with different heights and mass eruption rates.





## 1 Introduction

Uncertainty is an inherent feature of atmospheric dispersion problems, both in terms of how an initial release of material into the atmosphere is specified and in modelling the subsequent evolution of that material. It is important for us to acknowledge these uncertainties and their influences, and to work towards improving their representation when developing and applying our models of atmospheric dispersion processes. The uncertainty is introduced partly by our incomplete knowledge or understanding of a problem (a lack of information) but also there is an irreducible uncertainty arising from the stochastic nature of many physical processes. This stochastic uncertainty occurs at the small turbulent scales not explicitly resolved by our modelling system, but it also emerges at larger scales due to the chaotic nature of the evolution of our atmosphere.

The origin of uncertainties when modelling atmospheric dispersion for emergency preparedness and response applications (Leadbetter et al., 2020; Korsakissok et al., 2020; Sørensen et al., 2020; Le et al., 2021) can be grouped into three main categories. Firstly, there is uncertainty in the description of the source terms (Sørensen et al., 2019; Dioguardi et al., 2020), which may include aspects such as the release location, height, timing, species and quantities released, particle sizes, etc. This type of uncertainty can be particularly acute during the early stages of an incident when little may be known about the event but, as the event unfolds and more detailed information becomes available, improving knowledge can help to reduce this uncertainty. The second category is uncertainty in the meteorology, especially when forecasts are being produced for several days into the future. This is the focus of the current paper. Use of ensemble forecasting techniques has become the standard approach for representing uncertainty in weather forecasting (Buizza and Palmer, 1995) and is increasingly being applied in other fields where there is a dependence on the meteorology, such as for flood forecasting (Cloke and Pappenberger, 2009; Golding et al., 2016; Arnal et al., 2020) or forecasting of coastal storm surges (Flowerdew et al., 2009). Thirdly, consideration needs to be given to the limitations and approximations used within the dispersion model itself, including the choice of parametrisation schemes and values for internal model parameters. Model sensitivity studies can help explore this parameter space and better understand behavioural characteristics of the model (Harvey et al., 2018; Leadbetter et al., 2015; Dacre et al., 2020). Going beyond the dispersion modelling, there will, of course, also be uncertainties when assessing downstream 'impacts', e.g., when performing dose assessments or evaluating potential counter-measures for a radiological incident.

The relative importance of the three types of uncertainty will vary depending on the particular circumstances of an incident. On occasions, large uncertainties may exist in initial source estimates and will dominate the dispersion model predictions, while at other times, better constrained estimates of the source term may be available. The degree of uncertainty in the meteorological forecast will vary from day to day and from place to place, depending on the synoptic weather situation. Sometimes there is high confidence in the forecast evolution for several days ahead, while on other occasions forecast errors can grow rapidly in the first





day or two leading to a wide range of potential future weather patterns. As with the source term uncertainty, the meteorological
uncertainty is reduced as more information becomes available in the form of updated weather forecasts and eventually by the
use of 'analysed' meteorological fields. Potential errors arising from the dispersion model itself can be difficult to quantify
objectively during an incident. Model limitations can be explicit (where relevant processes are not represented in the model)
or implicit (where processes are represented but only in an approximate or limited way). It is important to consider whether
modelling options are appropriate for a given incident and any consequences such choices may incur.

Interactions between different types of uncertainty can also be important on occasions. For instance, uncertainty in release
time for a source could interact strongly with, say, meteorological uncertainty in the timing of a frontal passage. In a similar
vein, uncertainty in the height of a release can combine with uncertainty in wind shear behaviour related to the vertical wind
profile and it might determine whether a release occurs within the boundary layer or above it. These examples highlight
why a 'holistic' approach combining uncertainties is beneficial, though such an aim is also recognised to be quite a difficult
challenge in practice as some components of the dispersion problem might be poorly understood and their uncertainties poorly
constrained.

Various techniques are available to understand uncertainties in atmospheric dispersion outputs, with ensemble-based ap-
proaches becoming increasingly common over recent years (Sigg et al., 2018; Zidikheri et al., 2018; Maurer et al., 2021). In
weather forecasting, ensemble prediction systems first became operational in the early 1990s, with pioneering global ensemble
forecasts issued by the European Centre for Medium-Range Weather Forecasts (ECMWF: Buizza and Palmer (1995); Buizza et
al. (2007)) and the National Centers for Environmental Prediction (NCEP: Toth and Kalnay (1993)). Over subsequent decades,
many other operational centres have developed and now routinely run global and high-resolution regional ensemble systems
(Met Office: Tennant and Beare (2014), Canadian Meteorological Centre: Buehner (2020)). The development of ensemble
forecasting represented a paradigm shift in numerical weather prediction (NWP), in which a range of possible future weather
scenarios is presented, not just a single realisation, giving users an assessment of confidence in forecasts and allowing them
to make informed decisions based on a range of possible outcomes including extremes. While uptake has been slower for
ensemble-based dispersion modelling, the approach has started to become more widely adopted over recent years, recognising
the fact that considering uncertainties in the modelling process helps to guard against making inappropriate decisions based on
a single erroneous deterministic prediction. A number of studies have been carried out using ensemble dispersion model sim-
ulations of specific events for post-event analysis (e.g.,Sørensen et al. (2016); Zidikheri et al. (2018); Crawford et al. (2022);
Folch et al. (2022)), but only a few centres currently have a capability to produce real-time ensemble-based dispersion forecasts
in an operational environment (Dare et al., 2016).

A difficulty with validating dispersion simulations is that we often have little or no ground truth to verify against. Real-
world data involving atmospheric releases, either accidental or in planned dispersion experiments, are limited and observations
will vary in their types and quality. For this reason, atmospheric dispersion predictions are often evaluated against a model
simulation using 'analysis' meteorology, as is a standard practice in the verification of numerical weather prediction forecasts
(WMO (2019, Appendix 2.2.34), Hotta et al. (2023)). While there may be concerns with the approach not representing true
reality, it does allow sampling over a broad range and variability in weather situations and in a manner that the dispersion model





responds to, and is sensitive to, these different conditions (e.g., due to the changing influence of met variables in the dispersion model parameterisations). Building up verification measures over a large sample of events is also especially important when verifying probabilistic forecasts due to the extra 'dimension' of the probability threshold appearing in the analysis.

The aim in this paper is to explore meteorological forecast uncertainty propagated from an ensemble prediction system through an atmospheric dispersion model, with a particular focus on investigating the spread and calibration of the resulting ensemble of atmospheric dispersion predictions. We use the dispersion model NAME (Numerical Atmospheric-dispersion

Modelling Environment; Jones et al. (2007)) with ensemble meteorological forecasts taken from the global configuration of the MOGREPS (Met Office Global and Regional Ensemble Prediction System, Tennant and Beare (2014)) operational weather forecasting system. The paper builds on results presented in Leadbetter et al. (2022), which included an assessment of the skill of these ensemble dispersion predictions relative to deterministic forecasts using the Brier skill score. It was shown that the ensemble dispersion forecasts are, on average, more skilful than a single dispersion prediction based on a deterministic

meteorological forecast, although there are still occasions when the single deterministic forecast can give a better prediction. The relative benefit of the ensemble predictions over deterministic forecasts, as indicated through the Brier skill score, was also shown to be larger at later forecast time steps. In the present paper, these ensemble dispersion predictions are analysed in a different manner to assess how well the uncertainty in meteorology is captured in terms of the spread created in the atmospheric dispersion ensemble realisations, and to explore how well calibrated are our probabilistic predictions based on these dispersion

ensembles. Le et al. (2021) and Ulimoen et al. (2022) consider similar aspects of dispersion ensemble predictions in the context of the Fukushima Daiichi nuclear power plant accident in 2011, but compare their model predictions against measurements recorded at monitoring stations and subsequent survey data. Meanwhile, ElOuartassy et al. (2022) evaluates the performance of short-range ensemble dispersion simulations using $^{85}$Kr measurements recorded downwind of a nuclear fuel reprocessing plant during a two-month field campaign.

Two synthetic case studies involving hypothetical releases of material into the atmosphere are investigated: a near-surface release and an upper-air release. We have then chosen a nuclear power plant accident scenario and a large volcanic eruption as specific examples that fulfil these two criteria. As these are hypothetical events, there are no corresponding observational data available for forecast validation, and, as noted above, the choice can be made to validate predictions against equivalent model simulations produced using 'analysis' meteorological fields. While this synthetic approach is driven by a general lack

of suitable observational data sets for release events of this type, the method does facilitate analysis of a large collection of forecasts, which as already mentioned, is important for a robust verification of these ensemble forecasts.

Section 2 discusses the methodology and experiment design, and introduces the atmospheric dispersion model, NAME, and the ensemble NWP system, MOGREPS-G, used in this study. This section also describes the verification methods and metrics that are used to assess the performance of the ensemble simulations. Results for the radiological and volcanic ash case studies

are presented in Section 3 and are discussed further in Section 4. Finally, a summary and some concluding remarks are given in Section 5.



## 2   Method

This section introduces the modelling systems used in the study, and the experiment design and methodology. The verification methods used to assess ensemble performance will also be described. Our methodology adopts a *verification by analysis*
approach, which as discussed above is a technique commonly used in the verification of NWP forecasts. In this method, model forecasts are evaluated against model analyses that incorporate later observational data and may be regarded as an optimal estimate of the atmospheric state from that model. Often the same model will be used to provide a verifying analysis, though one downside to this is that any model biases that might be present will be common to both the forecasts and the analyses. As an alternative, an independent model can be applied to supply the verifying analysis. For the dispersion modelling in this
study, we verify ensemble NAME predictions against NAME simulations generated retrospectively using an 'analysis' data set derived from the sequence of global deterministic forecasts as described below.

### 2.1   Models

#### 2.1.1   Atmospheric dispersion model – NAME

The *Numerical Atmospheric-dispersion Modelling Environment, NAME*, is the Met Office's atmospheric dispersion model
(Jones et al., 2007) designed to predict the atmospheric transport and deposition to the ground surface of both gaseous and particulate substances. Historically NAME has been a Lagrangian particle model, though recent versions also incorporate an Eulerian model capability (not used in the present study). The Lagrangian approach uses Monte Carlo random-walk techniques to represent turbulent transport of pollutants in the atmosphere. Processes such as dry and wet deposition, gravitational settling and radiological decay can be represented in the model.

NAME was originally developed as a nuclear accident model in response to the Chernobyl disaster, and it continues to have an important operational role within UK and international frameworks for responding to radiological incidents (Millington et al., 2019). It is also the operational dispersion model used by the London Volcanic Ash Advisory Centre (VAAC) in response to large-scale volcanic eruptions to provide guidance to the aviation industry on the location and concentration of volcanic ash clouds (Beckett et al., 2020). NAME also has a wide range of other uses in emergency-response and research applications,
including modelling of airborne spread of animal diseases and plant pathogens, haze modelling and wider air quality forecasting applications, and inverse modelling for source identification purposes.

#### 2.1.2   Numerical weather prediction model – MetUM

Meteorological input fields for NAME are taken from the Met Office's operational numerical weather prediction system, *MetUM* (Walters et al., 2019). For this study, meteorological data are provided by the global ensemble and global deterministic
model configurations, as described below.

Ensemble meteorological forecasts are taken from the global configuration of the Met Office Global and Regional Ensemble Prediction System (MOGREPS-G) – an ensemble data assimilation and forecasting system developed and run operationally at



the Met Office (Tennant and Beare, 2014). The MOGREPS-G global ensemble produces 18 forecast members at each cycle, and runs four times per day at 00 UTC, 06 UTC, 12 UTC and 18 UTC, although only the forecasts at 00 UTC and 12 UTC have

been used to generate dispersion ensembles for the purposes of this study. A horizontal grid spacing of 0.28125° latitude by 0.1875° longitude is used (approximately equivalent to a spatial resolution of 20 km at mid-latitudes) and there are 70 model levels in the vertical extending from the surface to a model top at 80 km (only the lowest 59 vertical levels extending to an approximate altitude of 30 km are used for the NAME simulations). The temporal resolution of meteorological fields produced for NAME is three hourly.

At the time of this study, initial conditions for the global ensemble were obtained from the global deterministic 4D-Var data assimilation system, with perturbations added to represent initial condition uncertainty using an Ensemble Transform Kalman Filter (ETKF) approach. Structural and sub-grid scale sources of model uncertainty were represented using stochastic physics schemes that perturb tendencies of model variables such as temperature. The ensemble is optimised for error growth at all forecast lead times, though is primarily designed for short to medium range forecasting applications. It is recognised that the

ETKF perturbation scheme in use at the time of this study tended to produce forecasts with too little spread (Inverarity et al., 2023), a common shortcoming often seen with ensemble prediction systems. The current paper explores the extent to which under-dispersion is seen in dispersion modelling applications of these ensemble forecasts.

Meteorological fields produced by the global deterministic configuration (Walters et al., 2019) of the MetUM model are also used in this study for the purposes of verifying the ensemble dispersion simulations. The global deterministic model uses the

same 70-level vertical level set as the global ensemble, but has a horizontal resolution that is twice that used for the global ensemble, with a grid spacing of 0.140625° latitude by 0.09375° longitude (equivalent to a spatial resolution of approximately 10 km at mid-latitudes). As with the ensemble, the deterministic model runs four times per day, producing two full forecasts (out to 168 h) at 00 UTC and 12 UTC and two short 'update' forecasts (out to 69 h) at 06 UTC and 18 UTC. For this study, these high-resolution global deterministic forecasts are only being used to provide an 'analysis' meteorological data set for

producing a verifying NAME simulation of each release event. This 'analysis' data set is constructed by stitching together a sequence of successive short-range forecasts comprising the first six hours of each global forecast cycle.

The simulated releases begin six hours after the nominal initialisation time of each ensemble meteorological forecast. This choice partly reflects the typical delay of several hours that exists in the availability of real-time forecasts from the nominal model initialisation time, but it was also chosen to avoid giving an unfair advantage to any dispersion simulations produced

with the deterministic forecasts in Leadbetter et al. (2022) as these would otherwise have been verified against simulations using the same meteorological fields.

## 2.2 Experiment design

The paper examines two atmospheric release scenarios: a radiological release in the boundary layer and a volcanic eruption emitting ash into the upper air. Brief details of the modelling set up are summarised below, see Leadbetter et al. (2022) for a

more comprehensive description of the experiment design.





### 2.2.1 Modelling configuration for radiological release

The first scenario considers an accidental release of radiological material from a nuclear power plant. For the purposes of this study, a simplified hypothetical release of 1 PBq (= $10^{15}$ Bq) of caesium-137 (Cs-137) is modelled. The source term releases material into the atmosphere uniformly between the ground level and an elevation of 50 metres at a constant rate over a 6

hour time period. The caesium-137 is carried on small (non-sedimenting) particles which are subject to dry and wet deposition processes and to radiological decay (though the half-life decay of 30 years is negligible on the timescales considered here). All simulations consider a maximum plume travel time of 48 hours from the start of the release, so as to consider the effects of meteorological uncertainty during an initial response to a radiological accident.

An extensive collection of NAME simulations are produced for 12 release sites across Europe (see Figure 1) over a five-

month period from 2 Nov 2018 to 1 April 2019, designed to sample a range of different geographical locations and meteorological scenarios. Note that these are not locations of known nuclear facilities but are instead chosen to represent various topographical situations that could be subject to coastal effects, flow channeling by terrain, etc., and to sample variations in European climatology. Some sites such as Mace Head, Ireland are usually well-exposed to synoptic-scale meteorological influences, whereas other sites may be more sheltered or might be expected to see more influence from regional effects. The

simulations were performed using the 00 UTC and 12 UTC ensemble forecasts for every day in the period of the experiment. Each release starts six hours after the meteorological forecast data initialisation time.

Computational constraints on compute resource and data storage have limited the study to a five-month period of forecasts for a small number of mid-latitude sites in the European region (though our data set is still very extensive when compared with any other ensemble dispersion study to date). As a technical note, even with the above constraints, the full study (radiological and

volcanic ash scenarios) involved over 20,000 individual NAME simulations, using over 15,000 node-hours of HPC compute and collectively producing over 4 TB of model output data. Ideally several years of simulations and at a wider range of locations (e.g., including the tropics) would be needed to effectively sample the variability in meteorological conditions more universally.

Model output requested for the radiological scenario includes 1-hour mean air concentration and 1-hour deposition for each hour of the simulation, and total time-integrated air concentration and total accumulated deposition at the end of the 48 hour

forecast period. All quantities are output on a regular latitude-longitude grid with a nominal grid resolution of 10 km and with air concentration averaged over the near-surface (0 - 100 m) layer.

### 2.2.2 Modelling configuration for volcanic ash release

The second scenario considers the case of a large volcanic eruption of an Icelandic volcano. Three hypothetical eruption scenarios are examined: an eruption of the Hekla volcano (19.67° W, 63.99° N), summit elevation 1490 m asl, releasing ash

to an altitude of 12 km and an eruption of the Öraefajökull volcano (16.65° W, 64.00° N), summit elevation 2010 m asl, with eruption altitudes of 12 km and 25 km (see Figure 1). These three different cases are designed to sample possible eruption scenarios, albeit in a very limited manner. The two 12 km scenarios will test that results are not overly sensitive to small differences between broadly similar eruptions (in this case, there is a c.150 km lateral offset between the two volcanoes and





their summit heights differ by c.500 m introducing a factor 2.5 difference in the mass of ash released). The third scenario then
represents a much larger eruption event. To represent a range of different meteorological conditions, NAME simulations were
produced for the three eruption scenarios every 12 hours over the period from 12 Nov 2018 to 1 April 2019 (though technical
issues resulted in the loss of some simulations on two days during the study period). As with the radiological study, each release
starts six hours after the meteorological forecast data initialisation time.

All simulations use a model configuration based on the operational set up used at the London Volcanic Ash Advisory Centre
(VAAC) (Beckett et al., 2020). Each simulation covers a 24 hour period from the start of the eruption, replicating the duration
of formal forecast products produced by the VAACs. The eruption source term is assumed to be constant throughout the model
run. Mass eruption rates are determined using a relationship proposed by Mastin et al. (2009) between the height of the eruption
column and the erupted mass flux, and with the additional assumption that 5 % of emitted ash is fine enough to remain in the
distal plume and be transported over long distances in the atmosphere. Consequently, each scenario uses a different mass
eruption rate (Hekla 12 km: $8.79 \times 10^{12}$ g h$^{-1}$; Öraefajökull 12 km: $3.56 \times 10^{12}$ g h$^{-1}$; Öraefajökull 25 km: $1.13 \times 10^{14}$ g
h$^{-1}$). In particular, the 25 km eruption releases an order of magnitude more ash than the two lower level eruptions.

Airborne ash concentrations, ash column loads and accumulated ash deposits were output at 3-hourly intervals on a regular
grid with spatial resolution 0.314° longitude by 0.180° latitude (equivalent to approximately 20 km). Ash concentrations
are output for three height levels in the atmosphere (FL000-200, FL200-350, FL350-550) by following the same processing
procedure as employed by the London VAAC (Webster et al., 2012). Here, ash concentrations are first averaged over thin
atmospheric layers of depth 25 FL (flight levels), equivalent to 2,500 feet in the standard ICAO atmosphere, and then the
maximum ash concentration is selected from the thin layers that make up each of the three thick layers. In other words, the
thick layer outputs represent the peak ash concentrations that might occur at some altitude within each of the three altitude
ranges.

**2.3   Verification methods and metrics**

Forecast verification is more subtle with probabilistic forecasts than with deterministic predictions, as any single forecast is
neither 'right' nor 'wrong' but has to be judged in the context of a wider collection of cases when similar forecasts are made.
The requirement for adequate sampling is also more costly for probabilistic forecasts as they have an extra 'dimension' of the
probability threshold, so their demands on sample size are typically greater than those for deterministic forecasts.
There are established methods for the verification of ensemble forecasts, see Wilks (2019), which, in the broadest sense,
assess compliance with the *ensemble consistency condition* that underpins use of the ensemble for probabilistic predictions.
These methods allow features of the ensemble performance, such as forecast calibration, to be explored and can be used to
identify problems in the predictions such as a lack of spread or the presence of systematic biases. There are, however, some
subtleties in applying the techniques to dispersing plumes, in particular, due to the compact nature of plumes from a localised
source. For instance, the question is raised of how to define the area over which contingency table metrics are evaluated,
because any verification measures that take account of 'correct rejections' will be sensitive to this choice of evaluation area.





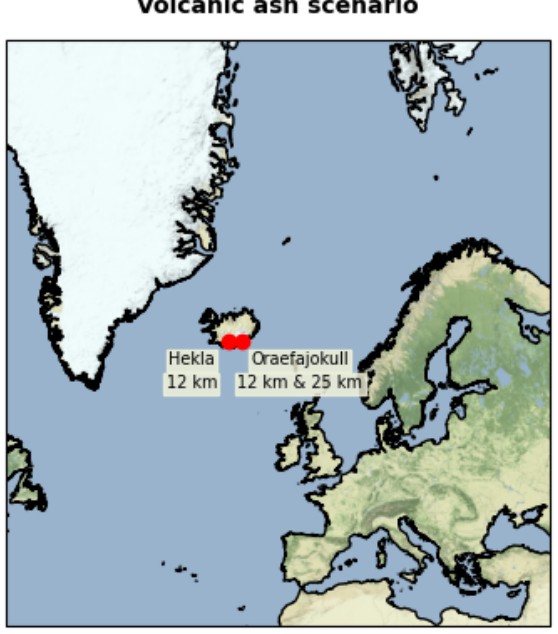

**Figure 1.** Geographical coverage of NAME output grids and locations for hypothetical releases: radiological case study (left) and volcanic ash case study (right).

Furthermore, 'climatology' (often used as a baseline against which skill is assessed for meteorological variables) is not a sensible concept for localised plumes of the type being considered in this paper.

Table 1 lists the output quantities from the NAME simulations together with threshold values used in their analysis. Data are sampled from the two-dimensional NAME output fields at a collection of 'virtual receptors' simulating a monitoring network.

### 2.3.1 The ensemble consistency condition

An ensemble forecast is said to satisfy the *consistency condition* if the true state of the system is statistically indistinguishable from the forecast ensemble, in other words, behaving like a random draw from the same distribution that produced the ensemble. In our case, the NAME simulation based on analysis meteorology is being used as a proxy for the true state. The extent to which the consistency condition is met by the ensemble will determine the quality of calibration of forecasts. For a consistent ensemble, probability forecasts based on the ensemble relative frequency will be well-calibrated, i.e., will accurately reflect





| Radiological scenario | |
| --- | --- |
| *Hourly output* | *Thresholds* |
| 1-hour mean air concentration | 1, 2, 5, 10, 20, 50, 100 Bq m$^{-3}$ |
| 1-hour time-integrated air concentration | 1, 2, 5, 10, 20, 50, 100 kBq s m$^{-3}$ |
| 1-hour deposition | 0.1, 0.2, 0.5, 1, 2, 5, 10 kBq m$^{-2}$ |
| *48-hour output* | *Thresholds* |
| Total time-integrated air concentration | 50, 100, 200, 500, 1000, 2000, 5000 kBq s m$^{-3}$ |
| Total accumulated deposition | 0.5, 1, 2, 5, 10, 20, 50 kBq m$^{-2}$ |

| Volcanic ash scenario | |
| --- | --- |
| *3-hour output* | *Thresholds* |
| Ash air concentration (at 3 levels) | 0.2, 0.5, 1, 2, 5, 10 mg m$^{-3}$ |
| Ash column load | 0.2, 0.5, 1, 2, 5, 10 g m$^{-2}$ |
| Accumulated ash deposition | 2, 5, 10, 20, 50, 100 g m$^{-2}$ |

**Table 1.** Quantities and thresholds for the radiological and volcanic ash case studies.

the underlying forecast uncertainties. In practice, ensemble consistency is usually degraded by the presence of biases and other forecast errors or by a misrepresentation of the ensemble spread.

### 2.3.2 Rank histogram

The *rank histogram* is a visual representation of the position of observed values within the ensemble distribution, and is a common tool for evaluating compliance with the consistency condition. The histogram is constructed by identifying where each observed value is located amongst the ordered collection of ensemble members (i.e., its 'rank') and then plotting these over many forecast instances to show the degree to which ranks are uniformly distributed. For an ensemble satisfying the consistency condition, the observation may be considered as an equally probable member of the ensemble and so occurs, on

average, at any rank, resulting in a 'flat' rank histogram. Deviations from rank uniformity indicate non-compliance with the consistency condition and the nature of these deviations can help to diagnose ensemble deficiencies. For instance, ensembles that have incorrect spread, but are otherwise broadly unbiased, can be identified by anomalous use of the most extreme ranks. Typically, ensemble forecasts tend to be under-spread, where the observed value is too frequently outside the range of the ensemble, leading to a characteristic 'U'-shaped rank histogram.

In practice, there can be some subtleties in how the rank of an observation is determined, e.g., when its value matches one (or multiple) ensemble members. This is especially pertinent for dispersion ensembles where the observed value might often be zero and some of the ensemble members are also zero. In this situation, a fractional value for the rank is assigned to each of the matching lowest positions in the rank histogram (for example, when the analysed value at a point is zero and there are 3 ensemble members also predicting zero at that point, a rank value of 0.25 would be assigned to each of the 4 lowest rank





categories). Consequently, for an under-spread ensemble, the over-population in the lower tail can effectively be spread out and affects the neighbouring bins, whereas at the same time there is a relative deficit in the population of the upper ranks (excluding the upper tail itself). This can lead to asymmetric behaviour between the lower and upper tails and also introduce an apparent downwards slope (from left to right) in the rank histograms. While it might be tempting to assign these data points exclusively against the lower tail, or to just ignore them entirely, such approaches would in themselves introduce biases and our approach

of using fractional ranks is an established method of handling situations when there are multiple ensemble members matching the observed outcome (Wilks, 2019). A novel approach of plotting the spatial distribution of rank contributions in a *rank map* helps to illustrate the above points.

### 2.3.3   Spread-error relationship

Rank histograms offer insight into how well the spread of the ensemble forecast matches with observed or analysed outcomes

and, in particular, can be useful in diagnosing over-confidence (too little spread) or under-confidence (too much spread) in the forecast. Further insight can be gained by looking at the ensemble spread directly, and comparing this against the errors in the forecast. For any individual forecast, the absolute error in the ensemble mean forecast can be plotted against the standard deviation for that ensemble forecast. The forecast error is, of course, not expected to match the standard deviation in any single forecast realisation. Instead, it is the distribution of these errors over a large sample of similar forecasts that is of interest here.

A formal consequence of the consistency condition is that *the root-mean-square error of the ensemble-mean forecast should be equal to the ensemble spread (that is, the standard deviation of the ensemble members)*. For an unbiased ensemble, a desirable characteristic is therefore for the predicted ensemble spread averaged over a set of similar forecast instances to be comparable to the average error seen in the ensemble-mean forecast on these instances. The degree to which this happens can be assessed by plotting the ensemble-mean forecast error against ensemble spread, after binning data points based on their forecast standard

deviation. An ensemble forecast that has the correct spread will lie along the 1-1 line, whereas a more typical under-spread forecast will be above the 1-1 line.

In practice, forecast biases will be present in any ensemble, though it is not straightforward to make any simple statement about them in the same way that we can say ensembles are typically under-spread. A forecast bias affects the spread-error relationship because the bias would increase the error but it would not be desirable to inflate the ensemble spread to match the

error as that can degrade the overall skill of forecasts.

### 2.3.4   Reliability (attribute) diagram

An important aspect of ensemble forecasts is that probabilistic predictions based on them are well-calibrated (i.e., forecast probabilities agree with the observed relative frequencies of events, with any deviations being consistent with sampling variability). Forecast calibration is illustrated using a *reliability diagram* (or *attribute diagram*). This has two components: the

*calibration function* showing the conditional relative frequency of events for each forecast probability category, and the *refinement distribution* showing the relative frequency of use of each forecast category and so the ability of the forecasts to discern different outcomes (a property known as 'sharpness').





The reliability diagram can highlight unconditional (systematic) and conditional biases that may exist in the calibration. A calibration function that is systematically above or below the 1-1 line reveals unconditional biases with under/over-forecasting of probabilities affecting all categories, whereas bias behaviour that varies across categories shows the presence of conditional biases in the probability forecasts. An under-spread ensemble forecast, that is otherwise unbiased, tends to give predictions that are too conservative (i.e., individual members lie towards the centre of the observed distribution) so that the frequency of forecasts involving the more extreme probability categories are misplaced, giving slopes flatter than the 1-1 line.

Historically, NWP ensemble forecasting systems have generally been found to be underdispersive and to have poor reliability (Buizza, 1995; Houtekamer et al., 1997), though modelling enhancements have steadily improved their performance over the years. The latest model configurations tend to be much better at representing ensemble spread and give reduced forecast errors and biases (Piccolo et al., 2019; Inverarity et al., 2023), but are not perfect, especially at local scales (Tennant, 2015). The spread and error characteristics can vary considerably depending on the parameter, and geographical and seasonal variations can also be large. The latter point argues the case for much longer ensemble dispersion studies to be carried out in the future, and at a broader range of locations around the globe, to better sample temporal and spatial variability in the performance of NWP ensembles.

## 3 Results

Results are first presented for the radiological case study and then for the volcanic ash case.

### 3.1 Radiological case study

We begin by considering the evolution of the hourly output fields for the radiological ensemble; in particular, looking at the 1-hour average air concentration and 1-hour deposition at six hour intervals over the course of the 48 hour simulations. While results vary to some degree between all release sites considered in the study, the two maritime sites at Mace Head and Biscay are more exposed to the background synoptic meteorology and tend to exhibit distinctive differences to the other sites. We will use Mace Head as our background site when presenting results to illustrate these differences.

#### 3.1.1 Rank histograms for hourly fields

Firstly, the degree of compliance with the ensemble consistency condition is assessed by examining their rank histograms. Figure 2 shows rank histograms that have been constructed using the combined dataset from all 12 release locations.

These plots show, for both quantities, the characteristic 'U'-shape distribution indicative of an under-spread ensemble where the analysed outcome is too often outside the range covered by the ensemble members. However, to put these plots into some context, it is useful to note here that our rank histograms (and reliability diagrams later on) generally compare quite favourably to those typically seen for NWP model performance, though as noted above, met model performance can vary considerably depending on the weather parameter, geographical region and season (see, e.g., Figure 6 in Tennant (2015) for an example of a rank histogram for 2-metre temperature forecasts, and Figure 10 in Piccolo et al. (2019) for reliability diagrams for probabilistic





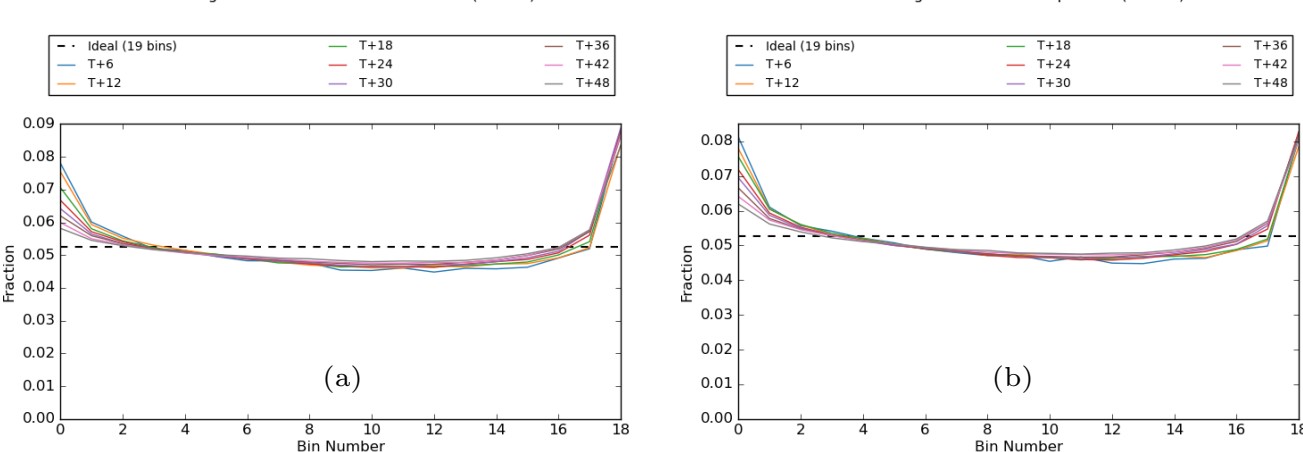

**Figure 2.** Rank histograms for (a) 1-hour averaged air concentration and (b) 1-hour deposition at 6 hour intervals from the start of the release. Plots are based on combined data for all 12 release locations. The ideal 'flat' histogram is shown by the dashed black line.

wind speed forecasts). This qualitative comparison with NWP ensembles is also consistent with the rank histograms shown
in Ulimoen et al. (2022) where ensemble spread is represented better for predictions of activity concentrations than for wind speed in the underlying meteorological ensemble.

While remaining under-spread throughout the period of the simulations, there is some improvement towards later forecast steps. This is generally seen in the lower tail only, with asymmetric behaviour emerging between the tails. In other words, the ensemble becomes less likely to collectively overestimate the analysed value, but there is no similar improvement in the
ensemble collectively underestimating the analysed value. A plausible explanation for some of this difference between the tails is linked to the fact that our quantities are bounded below by zero and, as time progresses, the ensemble range may be able to extend down to include zero but find it more difficult to extend upwards to encompass the analysed value due to the limited sample size of the ensemble. As discussed earlier, asymmetry between the tails is also, in part, an artefact of how the rank is determined when the analysed value is zero, which also explains the slight downwards slope from left to right in these rank
histograms.

Additional plots (not shown) suggest that over-population in the lower tail is dominated by regions where the ensemble members are in good agreement with themselves but there is an offset between this ensemble envelope and the analysed plume. However, there is no similar preferential bias affecting over-population in the upper tail, which appears to occur more generally and, as remarked, may be a consequence of insufficient sampling due to the limited size of the ensemble. This also
helps to interpret the asymmetric behaviour in the tails as the relative contribution of cases where ensemble agreement is high gradually falls away at later forecast times. When data points are filtered by the exceedance of a specified threshold (in terms of the exceedance being encountered by at least one ensemble member or the analysis simulation), the behaviour of the rank histograms is broadly similar across different thresholds, although results suggest a slight increase in asymmetry between the



lower and upper tails as the threshold value increases. This is consistent with expected behaviour as plumes at higher thresholds
cover smaller areas and the agreement between ensemble members is likely to fall away more rapidly at later forecast times.

Further insight into the points discussed above can be gained by plotting the spatial distribution of the rank structure for
plumes from individual release events in the form of *rank maps*, see Figure 3. These plots highlight the contributions to
rank histograms attributable to different regions in the comparison between the ensemble predictions and the analysed plume.
A dipole structure can sometimes be evident in these rank maps, indicating a systematic difference between the ensemble
forecast and the analysed plume, e.g., on occasions when all the ensemble members might be faster or slower than the analysis
in advecting the plume away from the source. The difference in horizontal resolution between the ensemble forecast and
analysis meteorology might be playing some part here in these discrepancies in plume evolution, for instance, in areas where
flow is being channelled by the terrain or in coastal regions where the land/sea boundary is being represented differently at
the two resolutions. However, there may also be occasions on which the analysis meteorology has 'flipped' from the earlier
forecast cycle to a new state.

The dipole appears as the dark blue region (which represents the upper tail of the rank histogram where the analysed value is
larger than all ensemble members) and the dark red region (which represents the lower tail of the histogram where the analysed
value is smaller than all ensemble members but the analysed plume is still present to some extent at this location). The green
regions in these plots illustrate where the analysed plume is absent but there are some members of the ensemble that predict
presence of material at that location. Some green regions would be expected, especially the darker green representing ensemble
'outliers', though large areas of lighter green, representing a significant proportion of ensemble members at a location, are
similar to the dark red in indicating a systematic discrepancy between the ensemble and the analysed plume. For green regions,
however, the attributed rank is distributed across the relevant lower bins rather than being assigned entirely to the lower tail
itself - and so large areas of lighter shades of green in these maps would be responsible for asymmetry between the tails and
appearance of the bias or slope in the rank histogram. If our 18-member ensemble satisfies the consistency condition, one
would expect, on average, around 1 in 19 locations being in each tail (i.e., a dark blue or dark red region).

The rank map gradually evolves through each simulation and there is a general signal for any red tail regions to gradually
disappear at later time steps (and more quickly than the blue regions which often persist). This is shown to some degree in
Figure 3 where the blue and red regions have similar extent early on, but the blue region tends to dominate by the end of the
run (in many cases the difference is much more pronounced than in this example, with only blue regions remaining at later
forecast steps). This reflects the asymmetry between the tails as already discussed above.

The rank histograms shown in Figure 2 were constructed using the combined data from all 12 release locations across Europe.
The extent of differences in behaviour between these locations can be assessed by examining releases from the individual sites
in isolation. Some variation in rank histograms is observed between different sites, primarily in the extent of the tails, and
these are most noticeable at the earlier forecast time steps, see supplementary figure S1. Well-exposed maritime sites, such
as Mace Head and Biscay, have flatter (i.e., better) histograms, whereas larger deviations are evident at other sites, such as
for the releases at Kristiansand, Karlsruhe and Milan. The reason for these differences is not entirely clear but part of the
explanation is thought to be the influence of terrain and small-scale complexity, with local and regional effects presenting a



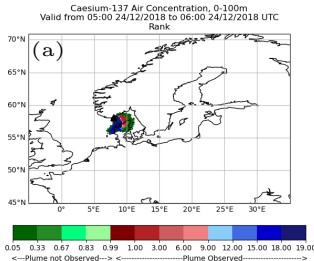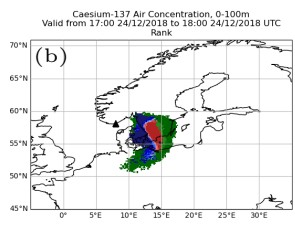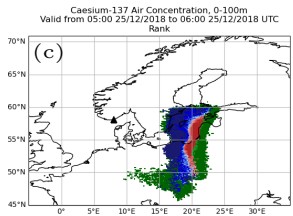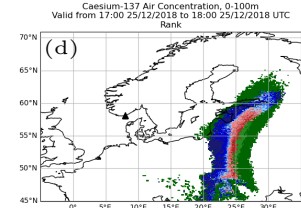

**Figure 3.** Rank maps illustrating contributions to the rank histogram from different regions of the plume released from Kristiansand at 18 UTC on 23/12/2018. At locations where the plume is 'observed' in the analysis, its rank within the ensemble distribution is plotted. Values for the rank range from 1 to 19 (the red-blue section). At locations where the plume is 'absent' in the analysis but predicted by some members of the ensemble, the proportion of ensemble members indicating the presence of a plume is plotted. Here, values range from 1-in-18 to 18-in-18 (the green section).

greater challenge to predict the plume evolution at some locations. Degradation in ensemble performance in regions with more

complex terrain is generally greatest during the early stage of a forecast and gradually improves at later time steps once the areal extent of the plume has grown. It is also possible that overall weather patterns are weaker and/or less predictable at some sites compared with the Atlantic and these climatological variations might also explain some of the differences that are seen. Systematic differences between sites become less obvious at later forecast steps, though it is notable that the two well-exposed maritime sites exhibit flatter histograms throughout the forecast period.

Rank histograms for 1-hour average air concentration at Mace Head (Ireland) and Kristiansand (Norway) are shown in Figure 4. The ability of the ensemble to capture (in an average sense over many realisations) the analysed outcome is generally very good at Mace Head. At T+6, and to a lesser extent at T+12, the rank histogram has a downwards trend from left to right, which as discussed earlier is actually indicative in this case of an over-confident forecast (related to the way that rank is determined when the analysed value is zero). The rank histogram indicates a slight tendency for the analysed value to be located

in the lower part of the ensemble distribution more frequently than in the upper part of the distribution, and conceptually, one could regard the situation where a significant number of ensemble members predict the presence of the plume in an area where it is absent in the analysis as being in some sense a case of the ensemble 'over-forecasting'. There continues to be a slight slope at later time steps too, but this is counteracted by the presence of an upper tail where the ensemble too often under-predicts the analysed value. The overall result is then a broad 'U'-shape pattern indicating the ensemble is very slightly over-confident

after the T+12 forecast step.

    The rank histogram for Kristiansand is more straightforward to interpret. One can again observe, on close inspection, a slight downward slope and an asymmetry between the left and right sides of the rank histogram. However, unlike at Mace Head, this is heavily outweighed at Kristiansand by the broad 'U'-shape pattern of an under-spread ensemble. The tails of the Kristiansand rank histograms are much larger than those for Mace Head, indicating the ensemble is over-confident. The

over-population in the tails reduces with increasing forecast time, especially in the lower tail, demonstrating that the ensemble





spread improves with forecast time. The reduction in the upper tail is more moderate, though still quite substantial over the period of the forecast, but the presence of this tail indicates that the analysed value is too frequently above the range of the ensemble.

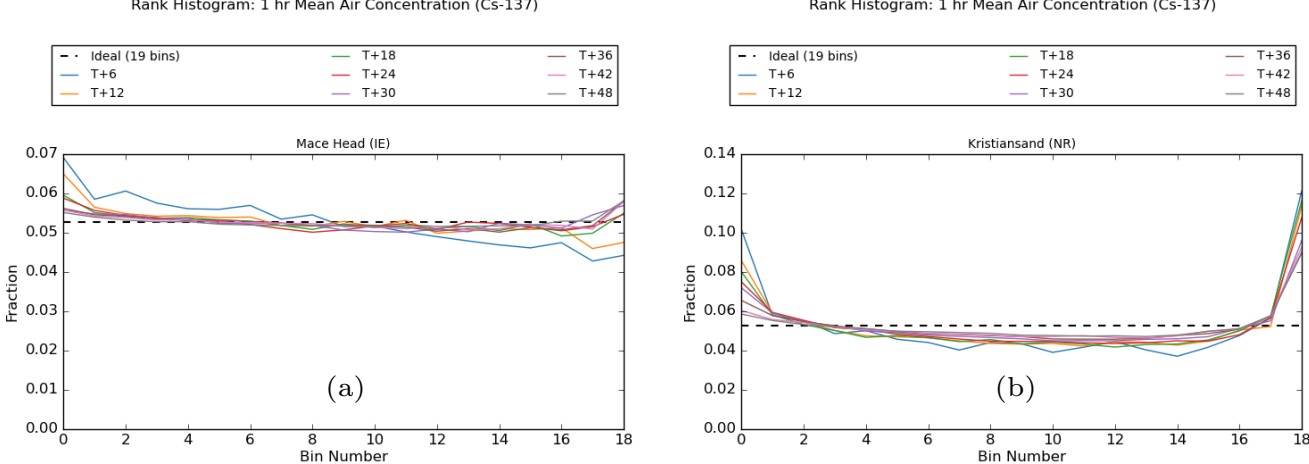

**Figure 4.** Rank histograms for 1-hour averaged air concentration at 6 hour intervals for (a) Mace Head and (b) Kristiansand.

### 3.1.2 Spread-error characteristics for hourly fields

Figure 5 shows a spread-versus-error comparison for the 1-hour averaged air concentration, plotted using the combined data from all 12 release locations. Plots on the left hand side show the absolute error in each individual ensemble-mean forecast plotted against the standard deviation for that ensemble forecast. If the consistency condition is satisfied, the root-mean-square error of the ensemble-mean over a large sample of similar forecasts is expected to match the predicted ensemble spread. As a visual guide, the 1-sigma and 2-sigma lines are also plotted, with approximately 68 % and 95 % of data points, respectively, 435 expected to lie under each line if we were to assume that errors here are Gaussian. However, it is difficult to judge the density distribution from the left hand plots alone and to establish the relationship more clearly an explicit binning of the data points based on the standard deviation of each ensemble forecast is shown in the right hand side plots. For a good ensemble forecast that has an accurate representation of ensemble spread, the root-mean-square error of the ensemble mean forecast is expected to approximately match the representative ensemble spread in each bin.

The upper plot shows the spread-error relationship using data points for all forecast time steps (at six hourly intervals). The scatter plot shows a clear trend for errors in the ensemble-mean forecast to be typically larger on occasions when the standard deviation of the ensemble forecast is larger. However, there is also a strong indication that errors can be substantially larger than the ensemble spread (in fractional terms) on occasions when the spread is predicted to be small, although the absolute magnitude of these errors are still generally small. When data is binned (right hand plot), it is apparent that the root-mean-445 square error in the ensemble mean is consistently larger than the ensemble spread across the entire range of spread values in





**Figure 5.** Comparison of ensemble spread against error in the ensemble mean forecast for 1-hour averaged air concentration at 6 hour intervals from the start of the release. In the upper plots (a), all time steps are combined, whereas lower plots (b) show the split by time step. Note the log scaling used in each plot, and the overall decrease in magnitudes as the time step increases due to progressive dilution of plumes. Plots are based on combined data for all 12 release locations.





these simulations, though the discrepancy is most noticeable in the lower bins and increases to one order of magnitude for the lowest bin range plotted. The poorer performance here might, in part, be attributable to regions near the edge of the ensemble envelope, especially when the analysed plume is an outlier. Results are qualitatively similar for the 1-hour deposition (see supplementary figure S2). For this quantity the predicted ensemble spread still tends to under-estimate the error, especially in

the lower bins again, but there is generally closer agreement with the error than in the case for the 1-hour air concentration. When comparing results from each release location (supplementary figures S3 and S4), the ensemble spread is closer to the error for air concentration at Mace Head and Biscay than at the other sites, though interestingly there is no obvious difference between sites for deposition.

In the lower plot, results are compared by the forecast time step. The left hand plot shows the same data as in the upper plot,

but with each data point coloured according to its forecast step. Data at later time steps are overplotted onto, and often obscure, existing data from the earlier time steps, but it does clearly illustrate a general progression to smaller concentration values as the forecast step increases and plume material is diluted. The right hand plot shows the binned data at each forecast step. Ensemble spread generally improves (relative to the forecast error) with increasing forecast time. This is especially evident for air concentration (as shown here). The representation of spread seems to be best in the higher bins that are captured at each

forecast time, with the ensemble tending to become increasingly under-spread in lower bins. Although somewhat less clear, there is a similar signal for deposition, but there is also a tendency for the ensemble forecasts to have too much spread for deposition in the highest bins at later forecast steps.

### 3.1.3 Attribute diagrams for hourly fields

Figure 6 presents an attribute diagram, also known as a reliability diagram, constructed from the combined data at all release

sites for various threshold exceedance values for the 1-hour mean air concentration at forecast step T+24. The calibration functions shown in these plots are flatter than the 1-1 line at all thresholds, which is a typical characteristic consistent with an over-confident ensemble forecast. This behaviour is also seen at other forecast steps (supplementary figure S5). Despite this shortcoming, the calibration of probability forecasts for threshold exceedances is generally quite good, especially at early steps in the forecast period and at small values for the exceedance threshold. There is a tendency to under-predict the frequency of

events occurring in the lower probability categories, and to over-predict their frequency in the upper categories, except at the highest thresholds where there is an indication of over-prediction of frequencies throughout the range. Also, the calibration skill falls off with increasing forecast step and this happens more rapidly at higher exceedance thresholds (supplementary figure S5). The refinement distributions, showing the frequency of use of the different forecast probability categories, demonstrate that these forecasts have good resolution and are able to distinguish forecasts at different probability values. Note the huge

peak in the number of forecasts where the probability of a threshold exceedance is assessed to be zero, which is associated with the large number of points in the model domain at which no material is present.

In examining the releases from individual sites (supplementary figure S6), there is some variation in the calibration performance between the different locations, but not excessively so. An interesting feature is the behaviour at Mace Head (and to a lesser extent at Biscay) where event frequencies in the lower probability categories are being slightly over-predicted at the 1.0





Bq m$^{-3}$ threshold, which is opposite to the behaviour seen at other sites. The reasons for this are unclear but could relate to these more exposed sites having generally higher wind speeds and narrower plumes (the resolution difference between ensemble and analysis meteorology fields may also be playing a role here in making it more challenging to 'hit' outlier solutions near to the edge of the ensemble envelope).

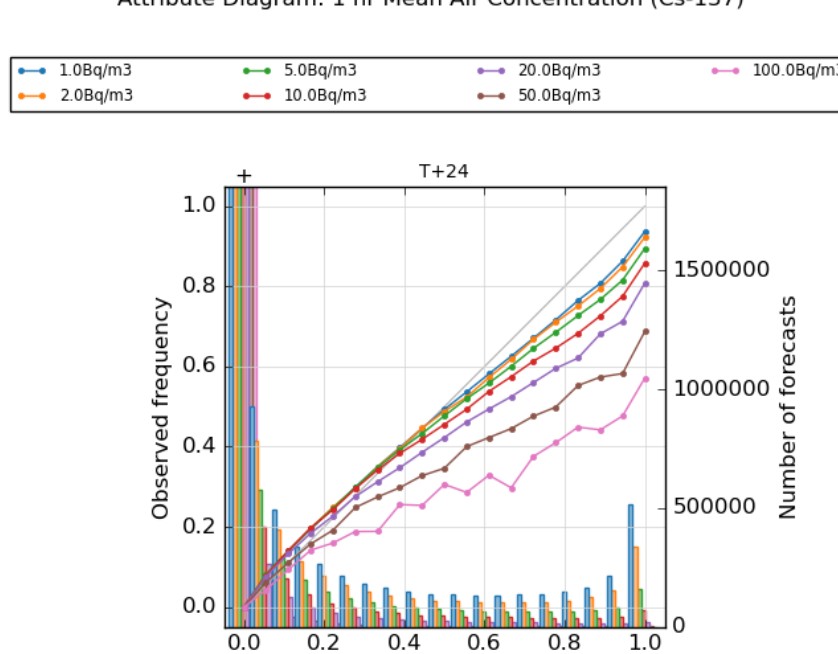

**Figure 6.** Attribute diagram for 1-hour averaged air concentration at 24 hours after the start of the release, evaluated for different concentration thresholds between 1.0 Bq m$^{-3}$ and 100.0 Bq m$^{-3}$. Plots are based on combined data for all 12 release locations.

### 3.1.4 Accumulated fields over 48 h period

The discussion up to this point has focussed on the time-step output (1-hour mean air concentration and 1-hour deposition). Quantities accumulated over the full simulation period are examined next; specifically, the 48-hour time-integrated air concentration and 48-hour accumulated deposition. Results for these accumulated quantities are broadly similar to the short-period averages, though some differences are also noted.

Rank histograms for the accumulated quantities, see Figure 7, show similar characteristics to those for the 1-hour average
quantities. Most notably, they are again 'U'-shaped in their appearance, indicating the under-spread nature of the ensemble predictions. It is perhaps not surprising that these rank histograms appear similar to an 'average' of the rank histograms for the 1-hour quantities (though formally the two operations of integrating fields over time and determination of rank are not




commutative). There could be occasions when time-averaging (in effect, averaging 'along' each plume) can help to mitigate forecast errors (for instance, the longitudinal averaging reduces the impact of any timing errors in the advection). However, more often than not there will be lateral errors in plume position where time-averaging is unable to help. The rank histograms indicate here that the length scale of the smoothing (i.e., plume extent) is shorter than the length scale associated with the ensemble error in plume position. While time-averaging would be expected to give smoother fields overall, it also tends to reduce ensemble spread so might not make it any easier for the analysed plume to be consistent with the ensemble distribution.

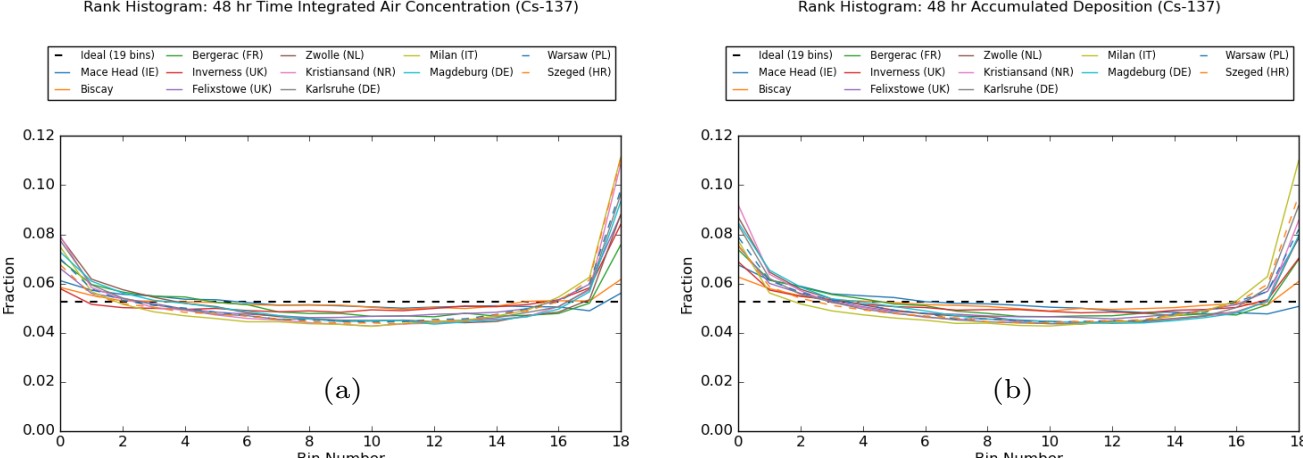

**Figure 7.** Rank histograms for (a) 48-hour time-integrated air concentration and (b) 48-hour accumulated deposition, plotted for the 12 separate release locations.

Comparing the rank histograms between the different release locations, relative performance is similar to that seen for the 1-hour average quantities. All sites exhibit some degree of being under-spread but the maritime sites dominated by synoptic-scale influences at Mace Head and Biscay have reasonably flat rank histograms compared with other sites where local and regional features have more influence and are better resolved by the higher resolution of the analysis meteorology. Further analysis (not shown) indicates the lower tail (ensemble over-predicts) again tends to be dominated by points where there is a systematic difference between the core of the ensemble envelope and the analysed plume, while the upper tail (ensemble under-predicts) appears to be more universally associated with points anywhere throughout the ensemble envelope. Additionally, when data are filtered by various thresholds to identify higher impact areas, the rank histograms appear broadly similar, though the upper tail shows a weak sensitivity to this threshold value (with larger tails evident at higher thresholds as accurately predicting the location of these high values becomes more challenging).

Spread-error comparisons, see Figure 8, are also very similar to the 1-hour average quantities with a clear relationship between the spread predicted by the ensemble forecast and the resulting error in the ensemble mean. When data are binned, the root-mean-square error in the ensemble mean forecast is consistently larger than the ensemble spread for all bins, more especially in the lower bins. However, agreement seems quite reasonable in the upper part of the range with only a small



under-estimation of forecast errors, and with the representation of spread being slightly better for accumulated deposition than for time-integrated air concentration (as also seen in the 1-hour outputs). For individual sites (not shown), there is generally closer agreement between spread and error at Mace Head and Biscay than at other sites for time-integrated air concentration, though differences between sites are less apparent for total deposition.

The calibration is generally good for ensemble forecasts of 48-hour time-integrated air concentration, as shown in the left hand plot in Figure 9. As with the 1-hour average air concentration, the forecasts under-predict the frequency of occurrence of low probability events and over-predict frequencies for the high probability bins, and consequently the calibration functions are generally flatter than the 1-1 line across most of the probability range. However, in contrast to the 1-hour air concentrations, the calibration function is now more symmetric in its appearance, with a cross-over between under and over prediction of probabilities occurring around or just below the probability value of 0.5. The extent of the under-prediction in the lower half of the range is also now broadly similar, though generally slightly smaller in absolute terms, than the degree of over-prediction seen in the upper half of the range. This over-prediction of high probabilities is most apparent at the higher threshold values, though overall the calibration is much less sensitive to threshold than for the 1-hour quantities. Therefore when comparing against the 1-hour quantities, calibration tends to be a little worse at low probabilities but better, and often substantially so, at higher probabilities.

When aggregated over all release locations, as shown in the figure, calibration errors can be up to approximately 0.04 (or ~20 % of the nominal probability) where errors are largest in the under-predicted (lower) half of the range, and up to approximately 0.12 (or ~12 % of the nominal value) in the over-predicted region at the top end of the range. When releases from individual locations are examined, some variation of the calibration functions is evident, and as seen with the other performance measures, sites such as Mace Head and Biscay tend to perform better on average across the range of probabilities. For example, comparing the calibration between Mace Head and Milan (supplementary figure S7), the largest absolute deviations from the 1-1 line (which tend to occur around the forecast probability value of 0.8) are 0.10 at Mace Head but are larger at around 0.20 at Milan.

For 48-hour accumulated deposition (right hand plot in Figure 9), the ensemble forecasts generally over-estimate event frequencies, but not massively so, and there is still a clear monotonic relationship between the predicted and observed frequencies, which would enable a simple recalibration of probabilities to be easily applied. Two regimes can be identified. At low probabilities (values up to around 0.2), calibration is very good, on average, but it is in this range that the calibration appears most sensitive to the threshold value, at least in fractional terms. Here the frequency of events is slightly under-estimated at low threshold values and over-estimated at higher threshold values but fractional differences are generally within approximately 10 % of the nominal probability (for aggregated data over all release locations). For values above 0.2, probabilities are consistently over-predicted with the largest calibration errors being approximately 16 % of the nominal probability, and while there is some small variation between calibration functions at the different thresholds, there is no indication of a systematic variation or sensitivity to the threshold.



(a)



(b)

**Figure 8.** Comparison of ensemble spread against error in the ensemble mean forecast for (a) 48-hour time-integrated air concentration and (b) 48-hour accumulated deposition. Plots are based on combined data for all 12 release locations.





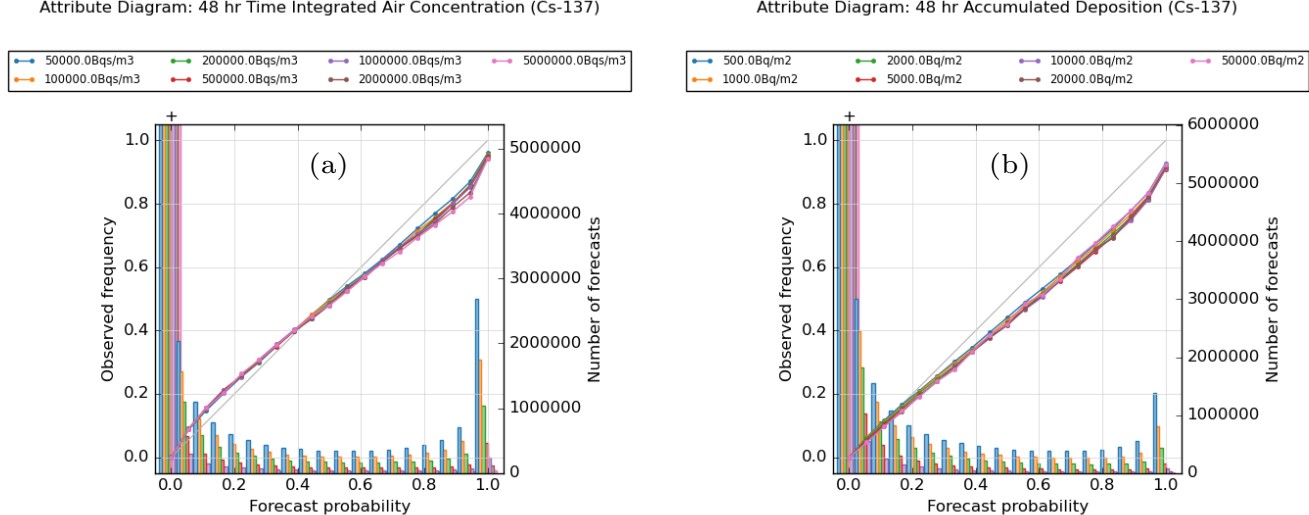

**Figure 9.** Attribute diagrams for (a) 48-hour time-integrated air concentration and (b) 48-hour accumulated deposition, evaluated at various thresholds. Plots are based on combined data for all 12 release locations.

## 3.2 Volcanic ash case study

For the volcanic ash ensemble, output at individual forecast steps consists of 3-hour averages for the ash concentration over three layers and the total column ash load, as well as the aggregated ash deposition up to that time in the simulation. These output fields are examined at 6-hour intervals from T+6 to T+24 to understand their temporal evolution through the course of the simulations. The total ash deposition aggregated over the full simulation period is also considered.

### 3.2.1 Rank histograms

Figure 10 shows rank histograms for the 3-hour mean peak ash concentration in each of the three standard VAAC flight-level layers, FL000-200, FL200-350 and FL350-550, with data combined from all three eruption scenarios. Compliance with the ensemble consistency condition appears to be generally good but characteristics vary by flight level and forecast step. The ensemble has an overall tendency to be somewhat under-spread, as indicated by the 'U'-shaped distributions, which is most evident in the upper-most level and improves with reducing height. The limited spread at higher altitudes is not entirely unexpected and is consistent with the model physics schemes being less active in the stratosphere as a mechanism to introduce forecast spread (for instance, there is no convection and no clouds) along with a gradual diminishing influence of observations with height in the ensemble data assimilation process. Conversely, the ensemble is slightly over-spread in the lower and middle layers at the first output step T+6, which might reflect a known limitation of the perturbation mechanism used in the ensemble NWP system at that time. Also, as seen in the radiological case, there is a tendency in the lower levels for a slight downwards trend from left to right at later forecast steps, which is thought to be attributable to the handling of points where there is a





mismatch between the ensemble forecast and the analysed plume and where the rank contribution gets distributed across a broader region than the lower tail (in other words, the slope is an indication of deficiency in ensemble spread rather than
565 systematic bias in ash concentration values).

Rank Histogram: 3 hr Ash Concentration (FL000-FL200)

Rank Histogram: 3 hr Ash Concentration (FL200-FL350)

Rank Histogram: 3 hr Ash Concentration (FL350-FL550)

**Figure 10.** Rank histograms for 3-hour mean peak ash concentration in the flight-level layers (a) FL000-200, (b) FL200-350 and (c) FL350-550, shown at 6 hour intervals from the start of the eruption to T+24. Plots are based on combined data from all three eruption scenarios.

The rank histograms for 3-hour mean ash column load at 6-hour intervals and the total deposited ash over the entire 24-hour period are shown in supplementary figure S8. The ash column load histograms show similar characteristics to those for the ash concentration in the middle flight level layer FL200-350, with the ensemble being initially over-spread at T+6 but otherwise the ensemble distribution is broadly consistent with the analysed outcomes. Meanwhile, the general appearance of the rank
histogram for the total ash deposits is very good, though the ensemble forecasts do appear to be slightly under-spread.




On examining the three volcanic eruption scenarios individually (supplementary figure S9), very similar results are observed for each of the ash source terms. The good agreement here is despite a considerable variation across these scenarios in the quantity of ash that is released into the atmosphere and, in the case of the Öraefajökull 25 km release, the height of the initial ash column. This would suggest that the rank histogram characteristics are attributable to the ability of the ensemble to capture the overall magnitude of the ash concentration at a location irrespective of its absolute value. Consistent with this idea, rank histograms are seen to be largely insensitive to the application of a threshold to filter the data points, at least for a wide range of threshold values of relevance to flight safety (ranging from 0.2 to 10.0 mg m$^{-3}$ for ash concentration and comparable ranges for the ash column load and ash deposition).

### 3.2.2 Spread-error characteristics

The characteristics of the rank histograms shown above would indicate that ensemble spread is generally well represented in the volcanic ash scenario, and this is thought to reflect the more constrained nature of the variability in the upper atmosphere in general. This is confirmed by quantitative assessment of the spread-error relationship for these ensemble forecasts. Figure 11 shows the spread-error characteristics for the 3-hour mean ash column load. The upper plot shows data combined from all three eruption scenarios. For the binned data, root-mean-square error tends to be slightly larger than predicted ensemble spread, except in the highest bin where the ensemble spread is marginally too large, but overall there is a reasonable match in the middle and higher bins (and agreement is generally better than that seen in the radiological case). The highest bin is populated by data associated with the Öraefajökull 25 km release scenario. When comparing results by each eruption scenario (lower plot), similar performance is observed in each case. The three eruption scenarios each release different quantities of ash into the atmosphere and, in particular, the Öraefajökull 25 km release is significantly larger than the two 12 km release events. These different mass eruption rates introduce a scaling offset in the plot, and after accounting for this, behaviour is very similar between the three scenarios. For each volcano, there is very good agreement in the highest bins relevant to that case (although ensemble spread becomes slightly too large for the Öraefajökull 25 km release as already noted), with the ensemble becoming progressively more under-spread relative to the forecast error in lower bins (though is still regarded as being reasonably good). Note that, because of the much greater quantity of ash released, the Öraefajökull 25 km case covers a broader range of values extending the data by almost a further order of magnitude over the 12 km releases. The poorer level of agreement between spread and error seen in lower bins, especially for the Öraefajökull 25 km data, might possibly be the influence of regions near the edge of the ensemble envelope, especially when the analysed plume is an outlier and forecast errors become large (in relative terms). Results are similar for the 3-hour mean peak ash concentration and also ash deposition, though ash deposition suffers from a lack of spread in general.

### 3.2.3 Attribute diagrams

Attribute diagrams for probabilistic forecasts based on the ensemble of volcanic ash predictions are shown in Figures 12 and 13. These attribute diagrams have been constructed using the combined data from all three volcanic eruption scenarios, though when examined separately (in supplementary material) the individual scenarios show broadly similar results highlighting that





**Figure 11.** Spread-error comparison for 3-hour averaged ash column load, using data evaluated at 6-hour intervals from the start of the eruption. In the upper plots (a), data for all eruption scenarios are combined, whereas the lower plots (b) show the split across the three different scenarios. Note the Öraefajökull 25 km eruption releases an order of magnitude more ash than the 12 km releases.





calibration does not appear to be particularly sensitive to the details of the volcanic eruption event. Probability forecasts for a
threshold exceedance are generally well-calibrated, especially later on in the forecast period. They also exhibit good resolution
with broad usage of the different forecast probability categories.

In Figure 12, calibration performance is shown for the 3-hour mean peak ash concentration in the lowest flight-level layer
FL000-200, calculated for an exceedance threshold of 0.2 mg m$^{-3}$. The ensemble is under-confident at early time steps but
this improves towards T+24 by which time it shows signs of becoming very slightly over-confident. Similar behaviour can
be seen for the two higher flight-level layers FL200-350 and FL350-550 (supplementary figure S10), where the degree of
over-confidence at T+24 becomes slightly more noticeable. In addition to the overall calibration being very good here, it is
also largely insensitive to the choice of threshold used for calculating exceedance probabilities (this is also seen with the ash
column load and ash deposition fields).

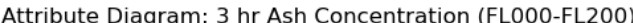

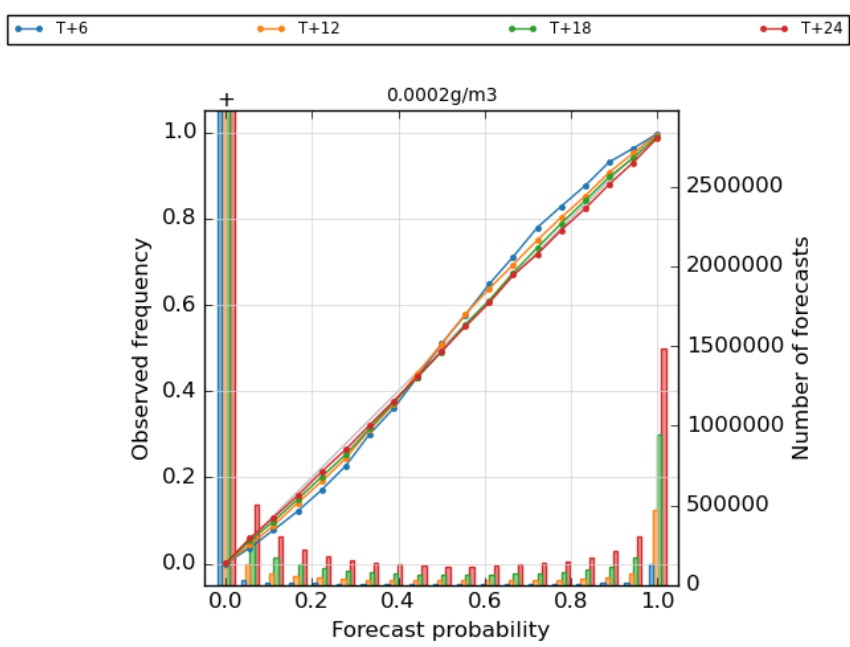

**Figure 12.** Attribute diagram for 3-hour mean peak ash concentration in the lower-most flight-level layer FL000-200, shown at 6 hour
intervals from the start of the eruption to T+24. Plots are based on the combined data from all 3 eruption scenarios and are evaluated using
an ash concentration threshold value of 0.2 mg m$^{-3}$.

Figure 13 shows forecast calibration for quantities at the end of the simulation period. For ash column load, there is a
small but noticeable over-prediction of probabilities in the lower and middle bin categories. The largest discrepancies are seen
around probability values of 0.3 where absolute probabilities are over-estimated by ∼0.05 or 15 % in fractional terms. On





examining the three eruption scenarios separately (supplementary figure S11), the over-prediction of probabilities in the low and middle bins is evident in all three cases but particularly so for the Öraefajökull 25 km release. Conversely, forecasts for the Öraefajökull 25 km release are well-calibrated for probability values of 0.6 and above, and help to partly offset a slight

tendency to over-forecast probabilities for the higher bins in the other two release scenarios. For the 24-hour accumulated ash deposition, calibration is excellent overall though there is a tendency to slightly over-forecast at the highest probabilities. On closer examination, this is dominated by the two 12 km releases, especially at the higher ash deposition thresholds shown here, while the calibration is better for the Öraefajökull 25 km release. The calibration functions generally appear somewhat noisier for deposition than for ash load (more noticeable when examining the individual scenarios) but this might be partly related to

our selection of thresholds leading to fewer sample points.

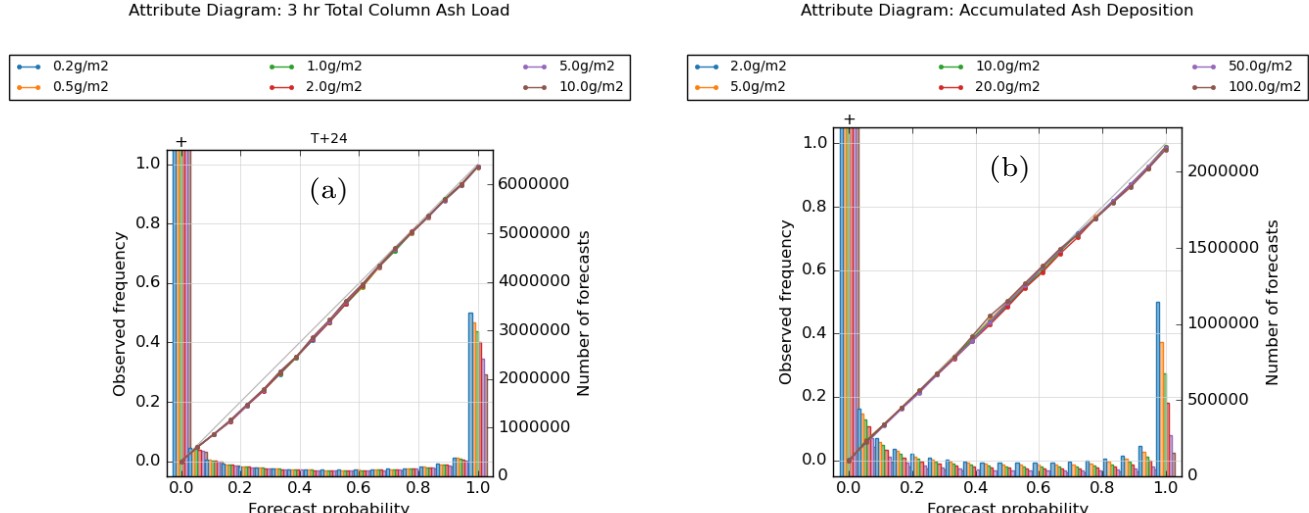

**Figure 13.** Attribute diagrams for (a) 3-hour mean ash load at T+24 and (b) accumulated ash deposition at T+24, evaluated at various thresholds. Plots are based on the combined data from all three eruption scenarios.

It is interesting that the calibration behaviour is different for the 3-hour mean total column ash load at T+24 from that seen for the 3-hour mean peak ash concentration on the three flight-level layers, in that there is a distinct over-forecasting of probabilities for ash load in the lower half of its calibration function, which is not seen for the ash concentrations. The reason for this is not entirely clear, although as the mismatch in calibration is most evident for the 25 km eruption scenario, it might

reflect poorer performance of the ensemble at higher altitudes in the stratosphere. Note that FL550 is at an approximate altitude of 16 km a.s.l. and when there is a significant proportion of ash emitted above this height it will contribute to the total column ash load but not to the ash concentrations. This explanation is also supported by the observation that ash deposition shows good calibration, which indicates that the representation of ash in the lower part of the atmosphere is being captured well by the ensemble forecasts. It is also possible that, because footprints are generally larger for total column ash than for ash in the

individual layers, the limited ensemble sample size and effect of outliers may be having more influence for ash column load.





## 4    Discussion

Ensemble NWP systems are carefully designed to represent uncertainty in meteorological forecasts. However, when used as input to a dispersion model, it is non-trivial how this uncertainty in the meteorological fields will propagate through to uncertainty in the dispersion predictions. This is because particle trajectories (and uncertainty in these trajectories) are integrated over time, which aggregates uncertainties and complicates the picture of relating them back purely to the synoptic variability at any point in space and time. Typically there are multiple time scales involved when modelling dispersing plumes, and whereas meteorologists usually only consider uncertainty in meteorological variables as a function of the forecast step, dispersion modellers might need to consider other relevant time scales such as the meteorological forecast lead time at the start of a release event, the release duration and the averaging or integrating period that might be appropriate when calculating concentration, deposition, etc. The two scenarios examined in this paper start to explore some of this complexity that links meteorological and dispersion uncertainties. The studies covered a period of several months and looked at various release locations and scenarios, though it is recognised the datasets could be too limited to explore detailed relationships, such as sensitivity to synoptic weather regimes or seasonal variations. The performance of the dispersion ensembles and their suitability for each application will now be judged.

### 4.1    Ensemble performance for radiological modelling

The results of the radiological case study indicate that there is a general overall tendency for too little spread amongst the members of the radiological dispersion ensemble. The under-spread, or over-confident, nature of the ensemble being demonstrated by the typical characteristics of U-shaped rank histograms, spread-error plots where forecast errors are systematically larger than the predicted ensemble spread, and calibration functions where the slope is generally flatter than the 1-1 line. For hourly output quantities that evolve with time, ensemble spread generally improves through the 48-hour simulation period in terms of gradually reducing tails in rank histograms and closer estimates, on average, to the errors seen in the ensemble-mean forecast. Overall performance is generally similar for the 48-hour time-integrated quantities often more relevant for decision making.

When looking at the plume predictions, there can be mismatches between the ensemble and the analysed outcome, particularly in the early stages after a release of material, giving regions where there is broad under-prediction and other regions of over-prediction (the 'tails' in the rank histogram). As plumes evolve, differences in behaviour emerge between the lower tail (where the ensemble members over-predict leading to false alarms) and the upper tail (where the ensemble under-predicts or misses an event). At early forecast steps, ensemble spread can be small and less likely to capture the analysed plume, giving larger tails on both sides. At later times, the range of ensemble solutions has broadened so that, at many points, the analysis is now captured within the range of members, giving a reduced lower tail, but the ensemble might still underestimate the magnitude of values at locations in the core of the analysed plume. This raises an interesting question on the influence of ensemble sample size, and whether the 18 members based on a single forecast cycle of MOGREPS-G are sufficient or would a larger ensemble be better able to resolve the peak values that might occur at any location? The ability of the ensemble to adequately sample these possible peak values would be important when making judgements on the areas at greatest risk from





a hazardous release. For weather forecasting applications, the Met Office uses time-lagging of two successive forecast cycles of MOGREPS-G to create a meteorological ensemble with 36 members.

From a decision making perspective, it is also encouraging that the best representation of ensemble spread (relative to forecast errors) is seen at the higher values of concentration and deposition, as these are likely to reflect the areas of most concern in any response. However, for deposition there is a hint of too much spread in the higher bins at later time steps (though otherwise the representation of ensemble spread is slightly better overall for deposition than for air concentration for both hourly and time-integrated quantities, though differences are small). More generally, and as alluded to above, the root-mean-square error in the ensemble mean forecast is typically larger than the ensemble spread (i.e., the ensemble has too little spread), and this is most evident for bins covering lower values of the spread. The poorer performance in these lower bins is possibly associated with regions near the edge of the ensemble envelope, especially when the analysed plume is seen as an outlier. However, as absolute values become quite small here, the consequences of the ensemble's inadequacy in capturing the spread are limited.

Another important aspect of the ensemble performance from a preparedness and response viewpoint is that probability forecasts based on the ensemble (e.g., for areas exceeding a specified threshold for air concentration or deposition) are well-calibrated, with observed frequencies of an event closely matching their predicted probabilities. The radiological ensemble demonstrates reasonable calibration overall. For the hourly air concentration output, there is a tendency to under-predict the frequency of events occurring in the lower probability categories, and to over-predict their frequency in the upper categories, except at the highest thresholds where there is a tendency to over-predict probabilities even in the low bins. The calibration deteriorates with increasing forecast step and this decline happens faster at higher exceedance thresholds as it becomes increasingly challenging to predict higher thresholds at the fine scale of the computational grid (this is similar to the problem of predicting high intensity rainfall in met ensembles). The limited size of the ensemble has an influence here. While calibration is sensitive to thresholds for hourly fields, it is much less so for the 48-hour time-integrated quantities and overall calibrations are reasonably good with estimated probabilities typically accurate to within around 10 % (though differences can be up to 20 % for releases at some sites). The good calibration for 48-hour quantities, and the fact that they are not particularly sensitive to thresholds, is important as these time-integrated quantities tend to be more relevant to decision makers. For 48-hour time-integrated air concentration, forecast probabilities tend to be under-estimated for the low probability categories (more so than for the 1-hour mean) and over-estimated in higher categories, whereas for 48-hour accumulated deposition, the forecasts more generally over-estimate the event frequencies across most of the probability range (but 48-hour deposition is often better calibrated than 48-hour time-integrated air concentration at low probabilities).

A notable feature in the radiological case is the variation in results between releases from different geographical locations. Rank histograms are flatter, indicating a better representation of ensemble spread, for release sites such as Mace Head and Biscay dominated by synoptic-scale influences, while performance is generally poorer at other sites, especially where it is thought there could be significant influence from terrain and other regional effects on the flow. Similar variation between release sites is seen when comparing ensemble spread against the forecast error, at least for air concentration where the comparison is again better for the Mace Head and Biscay sites, whereas inter-site differences are less obvious for deposition. When looking at the calibration of probabilistic forecasts of threshold exceedances, the picture also becomes less clear. However, there is





variation between sites and forecasts for the well-exposed background sites are generally better calibrated, particularly for the
48-hour time-integrated quantities. While it is useful to understand that ensemble performance can vary with geographical
location, it would not be straightforward to provide an a-priori objective assessment of the expected characteristics at any
specific location (and the performance at any individual site could itself vary with other factors through time such as the
synoptic weather pattern).

## 4.2  Ensemble performance for volcanic ash modelling

Results in the volcanic ash case study show ensemble forecasts generally performing well, with the spread of the ensemble
members capturing analysed plume values reasonably well and giving probability forecasts with good calibration. The volcanic
ash ensembles also tend to be more straightforward to interpret than in the radiological case, with performance that is largely
insensitive to the eruption scenario and impact threshold levels.

When looking at forecasts for peak ash concentration, the analysed outcome in the lower and middle flight levels is typically
not too different from the distribution predicted by the ensemble and their rank histograms are reasonably flat, though initially
the ensemble is slightly under-confident, i.e., has too much spread, for these levels. There are signs of an increasing tendency
for the ensemble to become over-confident in the upper levels, which would be consistent with the expectation of an increasing
lack of spread in MOGREPS-G forecasts at higher altitudes in the stratosphere as the influence of observations falls away and
as the less active model physics becomes less effective as a means for generating spread amongst the ensemble members. Good
quantitative agreement is also seen when comparing ensemble spread against forecast errors. The root-mean-square error in the
ensemble-mean forecast matches the ensemble spread reasonably well, though is often slightly larger. As in the radiological
case, it is encouraging that the agreement seems to be best at the higher concentration values that are of most relevance when
assessing impacts from an ash cloud. Results for ash column load and accumulated ash deposition are broadly similar to ash
concentration and again indicate a tendency for the ensemble to be initially slightly over-spread at T+6, while the performance
is slightly poorer overall for ash deposition with an indication that ensemble forecasts have insufficent spread.

Calibration of the volcanic ash probability forecasts is very good overall (and typically better than the calibration perfor-
mance seen in the radiological case study). For forecasts of threshold exceedance for peak ash concentration, probabilities
tend to be over-predicted slightly for those events associated with low probabilities but are under-predicted for events at high
probabilities. This is most noticeable at T+6, and is another indication that forecasts are under-confident at early time steps,
but improves towards T+24 by which time probability forecasts are generally well-calibrated though again with a hint that
forecasts have become slightly over-confident in the upper-most flight level. Forecasts for ash load and ash deposition are also
very well calibrated in general.

A notable feature of the volcanic ash ensembles is that results are similar for each of the three eruption scenarios considered
and are also largely insensitive to the choice of concentration thresholds, indicating that calibrations and other aspects of
ensemble performance do not seem to be particularly sensitive to the eruption height or the mass eruption rate. Peak ash
concentrations show strongest agreement here and differences becoming slightly more noticeable (though still small) for ash
column load and ash deposition. It is recognised that this case study only examined a very limited sample of three events from





which it might be difficult to draw firm universal conclusions and it would be beneficial to explore a wider range of releases as part of future work.

## 4.3 Comparison between the two case studies

Our two case studies examine the performance of ensemble dispersion forecasts in different parts of the atmosphere, with the radiological case primarily concerned with dispersion in the atmospheric boundary layer and the volcanic ash case focussing more on transport in the upper air. Apart from the height difference, there are other differences between the two modelling configurations that make it difficult to perform a like-for-like comparison between them. For instance, the length of the simulation is different between the two scenarios, and their quantities are evaluated using different averaging or integrating time periods and computed using different thresholds. Consequently it is not straightforward to disentangle differences that are attributable to the behaviour of the ensemble itself from those that arise due to differences in the modelling configurations, though results can be compared in a broader context to identify where there are consistent messages, as well as differences, between the two study cases. Our view is that most of the differences seen between the two cases are due more to the height of the release and its location rather than characteristics that are specific to each event type, such as physical form of the pollutant, particle sizes, half-life, etc. However, to understand the ensemble influence on the transport better in future work, a recommendation would be to model a generic release at different heights in the atmosphere from various locations with other aspects of the model configuration remaining fixed.

A common characteristic observed with NWP ensemble forecasting systems is their tendency to produce forecasts that have insufficient spread, with the observed outcome too often being outside the range covered by the ensemble members. It is therefore not that surprising that a similar lack of spread is seen in both of our experimental scenarios, but is particularly noticeable in the radiological study where ensemble dispersion forecasts are typically under-spread for all output quantities at all forecast time steps. For the volcanic ash study, forecasts are again often under-spread but characteristics vary by forecast step and by flight level, highlighting the three-dimensional nature of forecast skill. The spread is generally good in the FL000-200 and FL200-350 flight layers and, in fact, the ensemble shows signs of being over-spread here at the first forecast step T+6, but is under-spread in the uppermost FL350-550 layer. Spread is also generally well represented for the ash load and ash deposition quantities.

Probability forecasts based on the ensemble are reasonably well calibrated overall, though calibration of probabilities is generally poorer (though not necessarily poor) in the radiological study in comparison with the volcanic ash case, where it is often very good, especially after T+6.

Results in the radiological case show some sensitivity to the release location, which indicates a geographical variation in the skill of ensemble dispersion predictions. Better performance is observed at locations that are more exposed to the broader synoptic weather influence and less affected by local and regional influences such as the terrain. There is also likely to be some variation between locations due to their different climatologies and the skill of the forecasting system in representing such differences. Conversely, in the volcanic ash study, similar results are seen across the three different eruption scenarios considered, despite significant variation between scenarios in the height of the eruption column and the mass of ash emitted





into the atmosphere. However, all three eruptions are in the same geographical area and so wider variation in the meteorology is not captured in the same way as for the radiological study. Results are also largely insensitive to the choice of thresholds in the volcanic ash case, which is somewhat different to the radiological case where performance tends to fall off for higher threshold

events. This is an interesting difference for which the reasons are not entirely clear. It could, in part, relate to our choice of threshold values and their relative magnitudes compared with the amount of released material, such that the radiological case gets posed a greater challenge in matching predictions with the analysed plumes. The finer grids used for the radiological fields would also demand a more precise spatial match between fields, which could also help explain the degradation in calibration. Finally, the limited resolution of a global-scale NWP ensemble will find it intrinsically more challenging to represent near-

surface dispersion over that in the upper air, especially in regions where there is complex terrain and surface influences. The future use of high-resolution meteorological ensembles should provide a better representation of terrain and local effects, and might be expected to be beneficial for radiological applications and other near-surface releases.

### 4.4 Sensitivity to choice of verifying analysis

All results shown in this paper are based on verifying the ensemble forecast simulations against NAME runs using MetUM

global analyses. However, to assess sensitivity of results to this choice of the verifying analysis, verification has also been performed against NAME runs produced using global analyses from the European Centre for Medium-Range Weather Forecasts (ECMWF; Buizza et al. (2007)). Using an independent NWP model for the verifying analysis guards against the possibility that any systematic model errors in the MetUM could get concealed in the results, as such errors would affect both the forecasts and analysis in a similar way. While overall behaviour is broadly similar in qualitative terms, performance is generally worse when

verifying against ECMWF analyses, with more indication of over-confidence in the ensemble, particularly at early forecast steps, and also poorer calibration (as an example, supplementary figure S12 shows rank histograms and a reliability diagram constructed using the ECMWF analyses for the 1-hour air concentration forecasts in the radiological case study, equivalent to those shown in Figures 2 and 6 when evaluated using the MetUM global analyses). The poorer performance is not that surprising as ECMWF provides an assessment of the atmospheric state using an independent NWP model, whereas the MetUM

analysis shares some aspects of its base model and data assimilation scheme with the MOGREPS-G forecasting system and so is likely to provide solutions that are 'closer' to the ensemble forecast. In a sense, this emphasises that our assessment of ensemble performance in this paper only shows potential skill of these ensemble forecasts when other sources of uncertainty and errors, including in the underpinning NWP met model formulation, are ignored.

## 5 Summary and concluding remarks

The performance of an ensemble-based dispersion modelling system coupling the NAME dispersion model with MOGREPS-G global met ensemble forecasts is assessed using two case studies involving hypothetical atmospheric releases across Europe and the NE Atlantic. The results build on previous work presented in Leadbetter et al. (2022) showing that these ensemble dispersion forecasts are, on average, more skilful than a single dispersion prediction based on a deterministic meteorological



forecast. The present paper extends the analysis to investigate the spread and calibration of ensemble dispersion predictions
and looks at how well the uncertainty in meteorology is captured in terms of its effects on atmospheric dispersion. While
overall performance is generally good, there are some interesting differences between the two case studies, with generally
better results seen for volcanic ash (upper air) when compared with the radiological (near surface) case. These differences may
reflect a greater challenge in capturing uncertainty from surface and boundary layer influences, and in particular, in realising
sufficient spread of the ensemble for boundary layer plumes.

For the radiological case, the ensemble tends to be under-spread in general, especially at earlier forecast steps though there
is some improvement at later time steps. Results also indicate evidence of insufficient sampling by the ensemble members,
with the limited ensemble size affecting its ability to capture peak values or to adequately sample outlier regions. Probability
forecasts of threshold exceedances constructed using the ensemble show a reasonable level of calibration, though they tend to
be too keen on using the extreme forecast probabilities (i.e., are over-confident). The calibration also gradually falls away with
increasing forecast time. The radiological results reveal a sensitivity to the release location for near-surface releases, with better
performance seen at sites that are dominated by synoptic-scale meteorological influences and generally poorer performance at
other sites, especially where there is thought to be significant local influence from terrain effects. The spatial resolution of the
meteorological forecast system might be playing a role here in limiting the ability of the ensemble forecast to represent some
of these smaller scale influences.

In the volcanic ash study, where material is released in the upper air, the general performance of the ensemble predictions
is very good, and typically better than for the near-surface releases. For ash concentration forecasts, the ensemble provides a
good representation of the spread, particularly in the lower and middle flight levels, though there is slightly too much spread
(i.e., the ensemble is under-confident) for these levels at the initial time step T+6. This initial under-confidence is thought to be
attributable to a known deficiency in the ensemble perturbation scheme in use at the time of this study. Conversely, there is a
tendency towards over-confident forecasts in the upper level, which again is consistent with an expected decline in the spread
of MOGREPS-G forecasts at higher altitudes in the stratosphere. The calibration of probability forecasts for ash concentration
threshold exceedances is very good overall, and actually improves through the 24 hour period of these simulations, apart
from in the upper-most flight level where, as noted, forecasts become over-confident by the end of the period. Unlike in the
radiological case, the calibration and other aspects of ensemble performance are largely insensitive to the choice of exceedance
thresholds and are also broadly similar across the three eruption scenarios, which gives encouragement that these performance
characteristics might apply to a broad range of eruption scenarios having different heights and mass eruption rates. Results
for the ash column load and accumulated ash deposition show generally similar features to those discussed above for ash
concentration, though differences between the eruption scenarios are slightly more evident for these quantities.

An upgraded configuration of the MOGREPS-G ensemble forecasting system based on an 'ensemble of data assimilations'
(En-4dEnVar; Inverarity et al. (2023)) was introduced in late 2019. It implements a more sophisticated approach than the ETKF
scheme, which had a tendency to over-inflate the initial perturbations but still produce forecasts with too little spread at later
forecast times. The new configuration gives more realistic initial perturbations and improves ensemble spread at all forecast
times by adopting an 'additive inflation' scheme (in addition to stochastic physics) to better account for errors and biases in the



forecast model. Future work will explore the extent to which these improvements feed through to have beneficial impacts on the representation of meteorological uncertainty within our NAME dispersion ensembles. There is a recommendation for future studies such as this to assess forecasts over a much longer time period and to examine a broader range of locations around the globe in order to capture the geographical and seasonal variations in skill of the NWP ensemble system and how they relate to ensemble predictions of atmospheric dispersion (e.g., the influence of different climatology in regions other than Europe, or different processes in the tropics compared with mid-latitudes). Additionally, to understand more clearly the ensemble influence on the transport, it would be beneficial to adopt an experiment design that uses a single modelling configuration for generic releases at various heights in the atmosphere, rather than using application-specific model set ups as in the current study.

Use of time-lagged ensemble forecasts for dispersion applications will also be investigated to address the issue of limited ensemble size. Combining a sequence of atmospheric dispersion predictions based on successive forecast cycles of a deterministic meteorological model is a cost-effective method to build a basic dispersion ensemble, see, e.g., Vogel et al. (2014). The same principle can be applied to time-lag ensemble forecasts and is commonly used in weather forecasting applications. For example, the two most recent sets of ensemble members from MOGREPS-G are routinely combined to form a time-lagged ensemble with an effective sample size of 36 members (Inverarity et al., 2023). Time-lagging is not only a cost-effective way to increase ensemble sample size but also helps to reduce the impact of any correlated errors that may exist amongst ensemble members in the separate forecasts. However, older forecasts are less skilful and their inclusion in the time-lagged ensemble can degrade short-range forecasts, so a balance needs to be sought between the benefits of a larger sample size against their overall collective skill. There may also be scope for the recalibration of probability forecasts where there are known deficiencies in the raw forecast probabilities derived from the ensemble.

Finally, it would be useful to extend the range of verification methods, e.g., to better assess the utility of ensemble dispersion predictions using measures such as the Relative Operating Characteristic (ROC) and economic value of the forecasts. However, the verification of plumes from discrete sources of the type being considered in this paper poses some conceptual challenges for measures based on contingency table metrics ('hits', 'misses', etc.). In particular, there is a question around how a 'correct rejection' is defined. Dispersing plumes tend to be compact, localised structures (at least in the first day or two), which means that large areas of the modelling domain will always be clear of any contaminant. Verification scores that use correct rejections are therefore sensitive to domain size, and the challenge is to distinguish between plausible zeros (i.e., modelled zeros at points where material could conceivably reach within the time period but does not) from trivial zeros (points where material could never realistically reach).

*Code and data availability.* All the code used within this paper, the NAME dispersion model code, the statistical calculation code, and the plotting code is available under licence from the Met Office. Please contact the authors to request access. The meteorological data used to drive the dispersion model in real-time were not generated by this project and are not archived due to the huge volumes of data involved. Data produced during this work, output from the dispersion model and the results of the statistical calculations are available via Zenodo at https://doi.org/10.5281/zenodo.4770066 (Leadbetter and Jones, 2021).



*Author contributions.* The project was conceptualised and formal analysis was carried out by ARJ and SJL. The original draft was written by ARJ. ARJ, SJL and MCH contributed to the discussion of results and revision of the draft.

*Competing interests.* The authors declare that they have no conflict of interest.

*Acknowledgements.* The authors wish to acknowledge the assistance given by Met Office colleagues in the preparation of this paper. Firstly, we would like to thank Warren Tennant in our global NWP team for various discussions during which he contributed his thoughtful insights into the MOGREPS-G forecasting system, and Christopher Steele and Phil Gill in our verification team for help in using the VerPy verification tool to produce rank histograms and reliability diagrams plots. We also express our gratitude to Frances Beckett and Nina Kristiansen in the Atmospheric Dispersion and Air Quality team for their advice on setting up the volcanic eruption scenarios, and to Ben Devenish in the same
team for constructive comments on the draft manuscript.




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
