# Peer review of "Using synthetic case studies to explore the spread and calibration of ensemble atmospheric dispersion forecasts"

_EGUsphere, 2023_

## Author Response (AR1)

**Authors' response to referee comments and manuscript changes**

egusphere-2023-628

Title: Using synthetic case studies to explore the spread and calibration of ensemble atmospheric dispersion forecasts

Author(s): Andrew Richard Jones et al.

MS type: Research article

**1. Referee comments**

**1.1 Referee #1**

*General comments*

The paper deals with the problem of the propagation of meteorological forecast uncertainty through atmospheric dispersion model. The ensemble prediction system with 18 forecast members from the MOGREPS-G has been used for performing atmospheric dispersion simulations using NAME model, so the final output is in the form of the ensemble of atmospheric dispersion predictions. The investigation of the spread and calibration of this ensemble is one of the main purposes of this work. Two main hypothetical scenarios have been investigated: low elevated radiological release for selected 12 sites in Europe and high or even very high elevated 3 volcanic ash releases. Very extensive simulations for a period of 5 months with two releases daily for both scenarios have been performed. Finally, a huge set of data has been produced thus giving sound ground for any statistical analysis. The setup of such experiment is highly appreciated and can be considered as recommended for making deep analysis of the behaviour of any atmospheric dispersion ensemble system, in particular the ones used in operational mode. The final aim should be estimation of uncertainty of atmospheric dispersion modelling for various meteorological conditions. In this respect at some stage a comparison with other models and real measurements will be also necessary, but first proper calibration of the ensemble is one of the key factors, and this is why in the paper the authors concentrate on the analysis of the spread and calibration. However, it could be probably worth to put the work into a bit broader context, so the reader could better understand the whole process of uncertainty analysis and complexity of this problem, the more so a number of works have been already published aiming at the analysis of various types of ensembles, both from theoretical and practical points of view. It should also added that the added value of such extensive calculations producing large data, is such that various analyses can be performed, for example by comparing the results for different places or at different meteorological conditions.

*Specific comments*

1. One of the basic questions related to the presented methodology is whether 18 members is enough to produce sufficient statistics to cover interested range of possible results. It seems that there are situations when this is not the case, and the authors are aware that either more ensemble members would be needed or other models can be applied. ECMWF produces large forecast ensembling that can be used to drive atmospheric dispersion calculations, however it'd be very time consuming. The other possibility is to produce multi-model ensemble, which usually has bigger

spread than the ensemble based on one dispersion model. In fact there are many articles already published dealing with these issues.

2. Table 1 contains thresholds used for both scenarios. Obviously, in case of operational system, the best would be, when these thresholds reflect some criteria used operationally. For radiological scenario mostly doses are applied in various criteria, however in some countries, like Austria also time integrated concentration and deposition are used. For example some agriculture countermeasures can be implemented, if time integrated concentration of Cs-137 exceeds 350 Bq*s/m3 or deposition is higher than 650 Bq/m2 (for iodine I-131 this is respectively 170 Bq*s/m3 and 700 Bq/m2). Thresholds shown in Table 1 are much higher, but this is obviously arbitrary choice of the modellers.

3. The authors use quite simple indicators (rank histogram, attribute diagram, spread-error relation), but it seems they are mostly sufficient. On the other hand it would be convenient to see the values in the form of table (ensemble spread vs error in ensemble mean) to see how the results are changing in time. Some additional indicators can be also considered: like factor of 2 for spread-error diagram.

4. The way of rank maps presentation with two colour sections is appreciated. However, the reader should be warned against too simple interpretation of these maps. The fact that the ensemble system predicts areas where "real plume" (i.e. from analysis) are not present does not mean that the ensemble gave bad prognosis. If the ensemble shows low probability for such areas it is fine, otherwise you can say that prognosis was not very accurate. The role of ensemble is to predict areas when plume can, but not necessarily, must appear.

*Technical corrections*

The main comment is related to the request of including mathematical formulas for quantities used in the article, firstly, in order to avoid any ambiguity, and secondly simply for the reader's convenience. This concerns also the way how the figures have been constructed.

**1.2 Referee #2**

*Synopsis:*

In this study, the performance of an ensemble of dispersion model forecasts based on an ensemble NWP system (MOGREPS) is evaluated. The ensemble forecasts are generated from hypothetical (radiological and volcanic) emissions at different locations within northwest Europe over a period of several months. The forecasted air concentrations and deposited masses from the ensemble system are evaluated against corresponding quantities obtained by running the dispersion model with a sequence of NWP analyses obtained from a high-resolution NWP model. The results indicate that the MOGREPS ensemble is generally under-spread near the surface and in the stratosphere. However, in the troposphere, the ensemble spread better matches the forecast error and the forecast probabilities appear to be well calibrated, matching observed frequencies quite well at lead times greater than about 6 h.

*General comments:*

I have no major criticisms about this study. The methodology appears to be sound, and the results are generally in line with expectations given the characteristics of the MOGREPS ensemble. The use of concentrations obtained from the use of "analysed" NWP fields (essentially the fields obtained

from NWP data assimilation) as "truth" is a good idea that averts the problem of finding high-quality observations of atmospheric pollutants such as volcanic ash in sufficient quantity, which can be a very difficult problem in practice. Having said that, verification against observed ash, even with limited data, would strengthen this paper. Another suggestion for the authors is to show a comparison of the ensemble mean RMSE and the control member RMSE scores as well corresponding RPS/CRPS values. This would better highlight the value of the ensemble approach over the deterministic approach and would enable the reader to judge whether the deficiencies of the ensemble near the surface or at high altitude are severe enough to make the additional computational cost of the ensemble unjustifiable for particular applications.

*Specific comments*

Line 6: "Performance of the ensemble predictions is measured against retrospective simulations using analysed meteorological fields". I think something like this would make the methodology clearer for readers not familiar with NWP jargon: "Performance of the ensemble predictions is measured against retrospective simulations using a sequence of meteorological fields analysed against observations".

Line 61: (related to comment above) This is an opportunity to clarify the meaning of "analysed" meteorological fields.

Figure 5(b): Clarify what " #points in bin" mean and fix the number layout if possible.

Figure 5(c): Clarify that colour scheme matches labels in 5(d).

Line 688: "met" -> "meteorological".

**2. Authors' response**

The authors thank our reviewers for their thoughtful and constructive comments, and we have addressed the points that have been raised in our revised manuscript where it is practicable to do so. Our detailed response to each review is given below.

**2.1 Response to referee #1**

Thank you Slawomir for your helpful review comments. We have revised our manuscript to take account of points that have been raised.

*General comments*

Addressing the general comments section, we have included several further general references on dispersion ensembles in an effort to put our work into a broader context. As the present paper is essentially a follow on to (Leadbetter et al., 2022) we did not wish to repeat too much background information in the introductory sections, but we agree that a little more context would help the reader here. We have also added a further sentence to emphasise the value of this extensive dataset for exploring various aspects of the ensemble behaviour (as suggested, e.g., to be able to compare results for different locations or meteorological conditions). However, we have caveated our statement as we do not have a sufficiently large modelling period or domain coverage to allow comprehensive analysis to be performed (ideally, one would wish to run such simulations sampling a period of several years and with global coverage!).

*Specific comments*

1. Yes, a single cycle of the MOGREPS-G forecast is limited to an ensemble size of 18 members. The Met Office does use a time-lagged approach (2 x 18 members) for other forecasting applications, which we plan to explore for dispersion ensembles too in the future – but this is beyond the scope of the current paper. We agree that ECMWF would also be an option – providing more members plus the benefit of an independent data assimilation system and NWP model. The multi-model approach is another possibility, of course. However, many of the multi-model ensemble papers have looked at ensembles where both the dispersion model and the meteorological model vary and our aim in this paper is to focus just on meteorological uncertainty and ensembles within the framework of a single dispersion model. Lots of work remains to be done around aspects such as optimal ensemble size, source term ensembles, multi-model or multi-parameter dispersion ensembles!

2. The selection of thresholds for the radiological case had been kept simple, with thresholds chosen to capture typical downwind distances for monitoring and detection in the event of a moderate radiological accident (i.e., up to several hundreds of kms) rather than using actual thresholds for, e.g., evacuation and sheltering decisions. Our source term has similarly been chosen in a largely arbitrary manner. While our choices are therefore somewhat arbitrary, they are pragmatic and useful. The volcanic ash case uses real thresholds as these are more clearly and uniquely defined in an international context. Our paper is not aiming to assess specific features like sheltering or food bans (with their specific requirements on quantities, averaging periods, thresholds, etc.) but to instead assess the dispersion behaviour in a general manner.

3. We have used metrics and diagrams that are commonly used when evaluating ensemble predictions and these should provide a good general overview of ensemble performance especially with respect to the spread and calibration characteristics. We feel that presenting a table of results in this instance could be over-simplistic and might be misleading on its own, whereas the graphics in our figures help to illustrate the level of variation that is seen with dispersion ensembles. We have added the FA2 (factor-2) shaded region on the (binned) spread-error plots to aid interpretation, though we have not evaluated the FA2 metric itself on the raw data points for the reasons outlined above.

4. We have revised the sentence starting on L385 to take account of the above. The previous text did cover some of these points (e.g., in the discussion around the 'dark' and 'light' green regions) but we agree that the discussion could be a little clearer here. Also, even in an instance when the forecast probabilities are high and the 'real plume' is absent, the ensemble forecast is not necessarily 'wrong' (as probabilistic forecasts should be validated over many forecast instances). As an aside, there is a broader challenge when evaluating metrics that use 'correct rejections' of how we define the area of plausibility for the comparison between forecasts and observed plumes (that is, how we differentiate between locations where a plume could plausibly reach within the time period of interest and the remote areas where it is impossible for the plume to reach and there are trivial null forecasts). This question of how to handle the 'zeroes' problem is mentioned briefly elsewhere in the manuscript.

*Technical corrections*

Formulae have been introduced to define spread and error quantities more clearly. We feel the existing descriptions on the construction of diagrams (in words) are sufficient and that introducing additional notation might not be helpful, but we have included a further reference to Wilks here.

**2.2 Response to referee #2**

We thank the reviewer for their positive comments on our work.

*General comments*

The reviewer has proposed various ideas for extending our analysis to enhance the content of the paper, and while we agree that these are all very good suggestions for future work, they would involve significant additional effort at this time and we would not wish to delay publication of the present manuscript. Addressing the suggestions individually:

a) *verification against observed ash for a real eruption event*
   We recognise that the current study is limited to hypothetical events and does not consider any observations of real-world events, which as the reviewer has noted tend to be very limited in practice. However, we would regard objective verification against observations as being a separate study and outside the scope of the current paper. We are pursuing separate research examining the behaviour of our ensemble forecasting system against recent eruption events although the work is not yet mature enough for publication.

b) *comparison of RMSE for ensemble-mean and control, and examination of RPS/CRPS scores*
   We agree that these are both very good ideas to examine ensemble performance in greater detail. However, our existing analysis toolkit does not currently provide all of these outputs and would need some further development. We would then need to re-run processing on the entire dataset, which requires significant computational effort and time. It is therefore not straightforward to address this point in the context of the current manuscript. However, our earlier paper (Leadbetter et al., 2022) based on this ensemble dataset has examined the relative benefits of the ensemble approach over a deterministic one through use of Brier score metrics which partly addresses the wider point being raised here.

*Specific comments*

Our revised manuscript has addressed all the specific comments raised in the review. We would like to thank the reviewer for highlighting these points and for their helpful suggestions to improve the manuscript.

Lines 6/61: we have clarified the wording around "analysed" meteorological fields as per the reviewer's suggested text.

Figure 5: the graphic has been updated to resolve the earlier issue of the overlapping bin size information. This has been achieved by adjusting the bin widths to make them slightly broader (and they are now more consistent with the other spread-error figures in the manuscript). The use of wider bins gives slightly smoother lines but does not change results in any material way. The text colour for the bin size values has also been changed from green to grey, and the colour for the T+48 data points changed from grey to olive green to improve visibility of the 1-1 line. A 'within-factor-of two' shaded region has also been included to aid interpretation. Additional text added to caption to address the other points raised.

Note that Figures 8 and 11 have also been updated with minor formatting changes for consistency with the revised Figure 5. We have also increased the line weighting in Figure 11 b) to improve their visibility.

Line 688: wording corrected.

**3. Authors' changes to manuscript**

Line numbers refer to revised manuscript.

L6: minor editing of text in response to referee #2 specific comment 1

L40: additional reference in response to referee #1 general comments

L63: minor editing of text in response to referee #2 specific comment 2

L71-72: minor edits to shorten the text

L75-83: new paragraph added to broaden discussion to mention multi-model ensembles, etc. in response to referee #1 general comments

L131-135: added sentence to emphasise value of our dataset in response to referee #1 general comments

L137-139: minor edits to shorten the text

L310-328: further discussion and formulae added in response to referee #1 technical corrections

L337: additional reference in response to referee #1 technical corrections

L408-413: revised text in response to referee #1 specific comment 4

Fig 5: various improvements to plotting and extra details added in caption (similar changes to Fig 8 and 11) in response to referee #1 specific comment 3 and referee #2 specific comments 3 and 4

Fig 6 caption: extra sentence included to aid interpretation of the diagram

L696-697: added sentence in response to referee #1 specific comment 1

L716: minor edit in response to referee #2 specific comment 5

L888-891: minor edit to shorten the text

L908: added acknowledgement to our reviewers

Added references: Draxler et al. (2015), Galmarini et al. (2004), Galmarini et al. (2010), Rao (2005) in response to referee #1 general comments